

# Dynamics of island mass effect. Part I: detecting the extent

Guillaume Bourdin[1*], Lee Karp-Boss[1], Fabien Lombard[2], Gabriel Gorsky[2], and Emmanuel Boss[1]

[1]School of Marine Sciences, University of Maine, Orono, USA
[2]Laboratoire d'Océanographie de Villefranche-Sur-Mer, Sorbonne Université, France

**Correspondence:** Guillaume Bourdin (guillaume.bourdin@maine.edu)

**Abstract.** In the vast Pacific Ocean, remote islands and atolls induce mesoscale and sub-mesoscale processes that significantly impact the surrounding oligotrophic ocean, collectively referred to as the Island Mass Effect (IME). These processes include nutrient upwelling and phytoplankton biomass enhancement around islands, creating spatial and temporal heterogeneity in biogeochemical properties. Previous algorithms developed for detecting IME using satellite data are based on monthly or longer averages of satellite derived chlorophyll concentrations. As such, they tend to underestimate the true extent of this phenomenon because they do not take into account sub-mesoscale and short term temporal variations and because of the sensitivity of the detection algorithm to single pixel variability. Here we present a new approach that enhances satellite data recovery by merging products from multiple sensors and applying the POLYMER atmospheric correction. By integrating modelled surface currents with higher temporal resolution satellite observations, we dynamically track chlorophyll enhancements associated with IME and the advection of detached patches and filaments over distances exceeding 1000 km from their source. Our findings, applied to four island groups in the South Pacific, suggest that the ecological influence of IME on the oligotrophic ocean is much larger than previously recognized. This work provides a foundation for improved mechanistic understanding of IME and suggests broader implications for ocean ecology in subtropical regions. The approach developed here could be also be applied in studies on biological responses to other mesoscale and sub-mesoscale processes in other parts of the world's oceans.

## 1 Introduction

The Pacific Ocean is the largest ocean on our planet, covering approximately one-third of Earth's surface. Embedded in this vast open ocean are remote islands and atolls that are a source of perturbations to the open ocean ecosystem. As winds and currents interact with island topography, they induce mesoscale processes (i.e. local upwelling, eddies) at their wake downstream of islands. These in turn alter vertical and horizontal fields of temperature, light, and nutrients (Eden and Timmermann, 2004; Dong et al., 2007; Hasegawa et al., 2009; De Falco et al., 2022, and references therein). In most cases, increased chlorophyll concentration ([Chla], see Table 1 for definitions of all acronyms and variables used in this manuscript) is observed in the vicinity of islands, likely triggered by nutrient inputs from land and/or upwelling of nutrient-rich deep water around islands (Shiozaki et al., 2014; Gove et al., 2016; Caputi et al., 2019). This phenomenon, known as Island Mass Effect (IME), alters the growth and mortality rates of plankton species and introduces spatio-temporal heterogeneity in biogeochemical properties in the surrounding oligotrophic ocean. Signatures and effects of these IMEs can be detected hundreds of kilometers away from islands around which they were initiated (Messié et al., 2020, 2022). The first study on IME evaluated the enhancement of





carbon fixation as a measure of productivity near Oahu island (Hawaii) relative to the background ocean (BO), which was defined as the furthest station along a transect (in that case, 30 km away from the island's shore). This approach assumed that the IME was confined to an area located between the island's shore and the location of "BO station" (Doty and Oguri, 1956). The first basin-scale study of IME used in situ chlorophyll fluorescence measurements (Dandonneau and Charpy, 1985) and showed ubiquitous enhancements of chlorophyll fluorescence in the vicinity of large islands in the western Pacific (e.g. Vanuatu, Fiji, Tonga and Samoa islands).

**Table 1.** Table of notation

| | |
|---|---|
| SPSG | South Pacific Subtropical Gyre |
| IME | Island Mass Effect |
| BO | Background Ocean |
| $IME_M$ | Island Mass Effect zone delineated with the Messié et al. (2022) algorithm |
| $IME_D$ | Dynamic Island Mass Effect zone delineated with the algorithm developed in this study |
| $IME_T$ | Total Island Mass Effect zone delineated with the algorithm developed in this study. $IME_M + IME_D = IME_T$ |
| $BO_M$ | Background ocean zone relative to $IME_M$ zone, defined as: $BO_M$ area == $IME_M$ area, located outside of $IME_M$ zone, and closest to the 30 m isobath |
| $BO_T$ | Background ocean zone relative to $IME_T$ zone, defined as: $BO_T$ area == $IME_T$ area, located outside of $IME_T$ zone, and closest to the 30 m isobath |
| $[Chla]$ | Total chlorophyll $a$ concentration ($mg.m^{-3}$) |
| $c_{p660}$ | Particulate beam attenuation coefficient at 660 nm ($m^{-1}$) |
| chl_min | Minimum $[Chla]$ detected in the first pixel band adjacent to the 30 m isobath (shallow pixel polygon) of each island |
| chl_max | Maximum $[Chla]$ detected in the first pixel band adjacent to the 30 m isobath (shallow pixel polygon) of each island |
| chl$_{5th}$ | $5^{th}$ percentile $[Chla]$ of the $IME_T$ predicted zone |
| chl$_{95th}$ | $95^{th}$ percentile $[Chla]$ of the $IME_T$ predicted zone |
| $\Delta[Chla]_{IME_T - BO_T}$ | $IME_T$ [Chla] enhancement computed as $[Chla]_{IME_T} - [Chla]_{BO_T}$ ($mg.m^{-3}$) |
| $\Delta[Chla]_{IME_M - BO_M}$ | $IME_M$ [Chla] enhancement computed as $[Chla]_{IME_M} - [Chla]_{BO_M}$ ($mg.m^{-3}$) |
| $\sum[Chla]_{IME_T}$ | $IME_T$ surface-integrated [Chla] ($mg.m^{-1}$) |
| $\sum[Chla]_{IME_M}$ | $IME_M$ surface-integrated [Chla] ($mg.m^{-1}$) |
| $\Delta\sum[Chla]_{IME_T - BO_T}$ | $IME_T$ surface-integrated [Chla] enhancement computed as $\sum[Chla]_{IME_T} - \sum[Chla]_{BO_T}$ ($mg.m^{-1}$) |
| $\Delta\sum[Chla]_{IME_M - BO_M}$ | $IME_M$ surface-integrated [Chla] enhancement computed as $\sum[Chla]_{IME_M} - \sum[Chla]_{BO_M}$ ($mg.m^{-1}$) |
| $SEM^f_{\Delta[Chla]_{IME_T - BO_T}}$ | Standard error of mean associated with $\sum[Chla]_{IME_T - BO_T}$ ($mg.m^{-3}$; see appendix B) |
| $SEM^f_{\Delta\sum[Chla]_{IME_T - BO_T}}$ | Standard error of mean associated with $\Delta\sum[Chla]_{IME_T - BO_T}$ ($mg.m^{-1}$; see appendix B) |



The limited accessibility to vast areas in the South Pacific Ocean make ocean color remote sensing approaches well-suited
for basin-scale studies of IME. Using long term averages of [Chla] from ocean color remote sensing data (July 2002 to June
2012), Gove et al. (2016) showed that IME is a nearly-ubiquitous phenomenon across the Pacific Ocean. They estimated the
magnitude of IMEs by looking at changes in [Chla] within a ∼30 km wide band around each island's 30 m isobath, relative to
BO reference pixels located just outside this band (Gove et al., 2013, 2016). In practice, this detection method uses the same
quantitative approach as Doty and Oguri (1956) and accurately assesses the magnitude of the [Chla] enhancement associated
with IME as long as the BO reference pixels are outside the region affected by IME. This assumption is reasonable for small
islands (most islands in Gove et al. (2016) were smaller than a 30 km equivalent spherical diameter) and when using multi-year
averages of [Chla] that tend to highlight only locations with permanent [Chla] enhancement (see below). A more recent basin-
scale study of IME aimed to capture more complex spatial heterogeneity around islands by defining a specific [Chla] contour
to delineate the extent of IME, allowing the detection of IME to extend further than 30 km away from the 30 m isobath (Messié
et al., 2022).

Generally speaking, approaches for the detection of IME from remotely sensed [Chla] require a full or nearly full pixel
data recovery over the entire study area for an accurate delineation of the extent of IME. Messié et al. (2022) used yearly and
monthly averages of 4 km spatial resolution [Chla] maps for their basin-scale estimation of IME. While this temporal and
spatial averaging enables the production of gap-less [Chla] maps, it reduces the ability to detect fine-scale heterogeneity in
space and time (Lee et al., 2018), only highlighting [Chla] enhancement observable at the same location over the time frame
of the averaging period and therefore generally confined to regions directly adjacent to islands. Indeed, determining the spatial
extent of the biological response of IME and its effect on the ecology and bio-geochemistry of the adjacent oligotrophic ocean is
challenging due to its spatial heterogeneity and the transient nature of phytoplankton responses to perturbations (Messié et al.,
2020; Cassianides et al., 2020). Surface ocean properties, as observed by satellite sensors, are advected by wind and currents
across a kilometer-wide pixel on a timescale of a few hours. Therefore, observations of the ocean using yearly averages only
capture spatial patterns due to dominant winds and currents over this time frame, ignoring spatial and temporal heterogeneity
caused by short-term wind and current variability. Thus, a more accurate quantification of IME extent and dynamics requires
temporal averaging of satellite data over shorter time scales (e.g. to resolve mesoscale variability up to two weeks) and tracking
the evolution of IMEs over space and time using surface currents data (Cassianides et al., 2020). Ideally, daily observations of
the entire global ocean would provide the necessary temporal resolution to track IMEs. In reality, satellite measurements of
the ocean surface in visible and near-infrared wavelengths are often obstructed by clouds or affected by sun-glint, limiting the
extent of data recovery at the necessary temporal scales.

Here, we present a method to increase satellite data recovery to improve spatial and temporal resolution of satellite observa-
tions by merging products from up to five different satellite sensors and using an atmospheric correction that is less sensitive to
glint and adjacency effect. These merged products reveal frequent occurrences of higher [Chla] patches that are detached from
islands and advected offshore (referred to as "delayed IME" in  Messié et al., 2020). The higher temporal resolution achieved
allows a more accurate estimation of [Chla] accumulation as a proxy for phytoplankton biomass accumulation (termed as
"blooms") associated with IMEs. Building upon the work of Messié et al. (2022), we integrate modelled surface currents to



develop a dynamic algorithm for the detection of IME. We applied this algorithm to four island groups in the South Pacific Ocean (i.e. Rapa Nui, Society Islands, Samoa, and Fiji) over a six-month period and show that accounting for detached patches significantly increases estimates of total [Chla] stocks associated with IME in the area of study. This implies that IME has a much larger impact on the oligotrophic ocean than previously estimated.

## 2 Method

### 2.1 Level-3 multi-satellite composites

The use of a single satellite sensor often results in maps with significant gaps in data due to intermittent cloud cover or glint (which depends on satellite-specific viewing angle). To address this, we have adapted NASA Ocean Color's processing strategy to produce level 3 custom-made composite products from level-1A (L1A) top-of-the-atmosphere radiance. We merged data collected by three different sensor types (MODIS, VIIRS, and OLCI) onboard up to five polar-orbiting satellites (Aqua, Terra, SNPP, JPSS1, Sentinel-3a, and 3b). By taking advantage of their different overpass times, swaths, and viewing geometry, we decreased the impact of clouds and glint on data recovery. Additionally, we applied the POLYMER atmospheric correction (Steinmetz et al., 2011) to further improve data recovery in areas impacted by glint and adjacency effect (e.g. close to shore and clouds, see Steinmetz et al., 2011). The conceptual diagram of the processing pipeline, from level-1 to level-3, is shown in Fig. A1.

#### 2.1.1 Level-3 satellite products computation

We downloaded all MODIS and VIIRS level-1 (L1A) images in the vicinity of islands of interest from the Ocean Color repository, and OLCI level-1 images from the Copernicus repository. We processed these L1A images into atmospherically corrected level-2 remote sensing reflectance ($R_{rs}$) data using the POLYMER algorithm. We removed bad quality data pixels by applying the flags and recommendations of POLYMER (Steinmetz et al., 2011). Subsequently, we projected each satellite image onto the same equally spaced one kilometer spatial resolution plate-carré reference grid using nearest-neighbor interpolation. We estimated [Chla] using the CI-OCx blended algorithm based on the most recent update of the color index algorithm (Hu et al., 2019) and the OCx algorithm (O'Reilly and Werdell, 2019). We computed surface-integrated [Chla] as a metric for two dimensional phytoplankton biomass in metric tons of Chla per depth meter ($mt.m^{-1}$) by summing the [Chla] of each pixel within a predefined zone (i.e. here, the zone influenced by IME) multiplied by the area of that pixel:

$$\sum [Chla]_{IME} = \sum_{n=1}^{N_{pixel_{IME}}} [Chla]_n \times area_{pixel_n} \qquad (1)$$

#### 2.1.2 In situ data

We calibrated remote sensing products to minimize inter-sensor variability and biases using in situ data collected during the *Tara* Pacific Expedition (Gorsky et al., 2019; Lombard et al., 2023). We measured hyper-spectral absorption ($a$) and attenuation





($c$) quasi-continuously near islands with a SeaBird ACs spectrophotometer mounted in an underway flow-through system. We computed particulate absorption and attenuation coefficients (i.e., $a_p$ and $c_p$) by referencing these sensor measurements to hourly samples taken through a $0.2\mu$m filter (Dall'Olmo et al., 2009; Slade et al., 2010; Boss et al., 2019). Particulate beam

attenuation at 660 nm ($c_{p660}$) was used as a proxy for particulate organic carbon (Gardner et al., 2006; Cetinić et al., 2012). We estimated absorption specific to [Chla]-containing particles using the line-height of the $a_p$ peak at 676 nm ($a_{p676LH}$; Boss et al., 2013). We collected surface samples daily around 10:30 am local time for pigment analysis via high-pressure liquid chromatography (HPLC; see Gorsky et al., 2019; Lombard et al., 2023). We then estimated [Chla] from $a_p$ by applying the well-constrained linear relationship between the logarithm of $a_{p676LH}$ amplitude and the logarithm of total [Chla] estimated

from HPLC (Fig. B2.a).

### 2.1.3 In situ and satellite match-ups

We performed match-ups between the calibrated [Chla] estimated from the underway system and the [Chla] estimated from satellites to choose the best algorithm (i.e. least noisy or biased) to compute [Chla] from satellite $R_{rs}$. We downloaded L1A top-of-the-atmosphere radiance from MODIS-Aqua, MODIS-Terra, VIIRS-SNPP, VIIRS-JPSS1, OLCI-S3a, and OLCI-S3b

along the entire *Tara* Pacific transect (May 2016 to October 2018 see Gorsky et al., 2019) with the python download utility "getOC" (https://github.com/OceanOptics/getOC) and processed them into atmospherically corrected level-2 $R_{rs}$. We then derived products following the same scheme as the level-3 products aforementioned but without re-projecting, nudging, or merging the products to keep each satellite's native resolution (Fig. A1). For comparison, we also generated the standard NASA $R_{rs}$ using the atmospheric correction of SeaDAS (i.e. "l2gen") using the Ocean Color processor (OCSSW) V2022.3.

We then estimated [Chla] from these $R_{rs}$ using the same blended CI-OCx algorithm (i.e. chlor_a; Hu et al., 2019) and the simple OCx algorithm (i.e. chl_ocx; O'Reilly and Werdell, 2019). We matched these three different [Chla] products (i.e. chlor_a_polymer, chl_ocx_seadas, chlor_a_seadas) with the calibrated [Chla] estimated from the underway system following Bailey and Werdell (2006). We extracted and averaged underway [Chla] measurements within a $\pm3$ hour period of each satellite overpass, and satellite data from the 25 closest pixels to underway data locations, following the application of rec-

ommended Level-2 masks. We computed median coefficients of variation of normalized water-leaving radiance ($nLw$) for bands between 412 and 555 nm and for the aerosol optical thickness at 865 nm for each match-up and tested several homogeneity thresholds and minimum number of unmasked pixels to maximize the number of valid match-ups without introducing noise to the in situ-satellite correlations (Bailey and Werdell, 2006). Only match-ups with a minimum of 7 unmasked pixels and coefficients of variation lower than 0.15 were kept (Fig. B2(b), (c), and (d)). We compared the parameters of the ro-

bust linear regressions of valid match-ups to choose for the best [Chla] derivation methods (Table B1). We found 33% more valid match-ups with [Chla] computed using POLYMER $R_{rs}$ (N = 428) than valid match-ups with [Chla] computed using SeaDAS $R_{rs}$ (N = 321). [Chla] computed with the blended CI-OCx using POLYMER $R_{rs}$ showed, on average, the highest coefficient of determination ($\overline{R^2}_{chlor\_a\_polymer} = 0.78 \pm 0.05$), slopes closest to 1 ($\overline{slope}_{chlor\_a\_polymer} = 0.99 \pm 0.10$), and intercepts closest to 0 ($\overline{intercept}_{chlor\_a\_polymer} = -0.06 \pm 0.10$) when compared to in situ [Chla]. In contrast, the normalized

root mean square error of the correlation between in situ [Chla] and [Chla] computed with the blended CI-OCx using POLY-



MER $R_{rs}$ ($\overline{nRMSE}_{chlor\_a\_polymer} = 21.81 \pm 6.34\%$) was higher than with the other two [Chla] computed using SeaDAS $R_{rs}$ ($\overline{nRMSE}_{chlor\_a\_seadas} = 16.55 \pm 1.70\%$ and $\overline{nRMSE}_{chl\_ocx\_seadas} = 20.94 \pm 3.18\%$). Considering the smaller bias (slope closer to 1 and intercept closer to 0) and better data recovery (higher number of valid match-up) associated with the computation of [Chla] with the blended CI-OCx algorithm applied on POLYMER $R_{rs}$, we choose this method for the rest of the analysis to minimize differences between sensors while maximizing valid pixel recovery.

### 2.1.4 Satellite products adjustment and merging

We followed a similar merging strategy to that of GlobColour: each sensor's satellite product was derived separately before merging them (Garnesson et al., 2019), rather than merging reflectances before calculating the products (Sathyendranath et al., 2019). This method offers two important advantages; (1) it does not require simulations of the 510 nm band, which are not available on VIIRS and MODIS, and (2) it benefits from sensor-specific algorithm coefficients that account for variability in $R_{rs}$ across sensors to produce consistent products (Garnesson et al., 2019). To improve consistency and minimize the differences across satellite sensors, we individually calibrated the [Chla] data from each sensor with the underway in situ [Chla] measurements (using parameters from their respective robust linear regressions, see Table B1) to produce "calibrated" products before merging them. This nudging method reduced the inter-satellite variability and improved the spatial smoothness of the binned products. Since [Chla] was calibrated to in situ data, the bias associated with the estimation of [Chla] from each satellite was centered, and likely reduced, to the bias of in situ data. For each study area, we binned the calibrated data temporally to reconstruct full satellite images. Time-series of 8-day medians were the smallest temporal binning we could achieve to recover nearly full satellite images in all the studied regions for six-month long time-series. Before computing the median of a given 8-day period and a given region, we grouped all re-projected level-2 images and removed outliers (see appendix C). We produced a six-month long time-series of 8-day medians of [Chla] for each of the four case-studies presented here. Each case-study region was centered geographically on an island sampled during the *Tara* Pacific Expedition, and each six-month time-series was centered temporally on the day of in situ sampling (Gorsky et al., 2019; Lombard et al., 2023). We propagated errors associated with [Chla] estimation, nudging, and merging throughout each step to represent the final [Chla] uncertainty of the merged product ($SEM^{f}_{[Chla]_{IME}}$; appendix B). We used this final uncertainty to determine if the [Chla] enhancement associated with an IME was significant or not.

### 2.1.5 Spatial resolution

Most operational level-3 products are available at spatial resolutions of 4 or 9 km. While this resolution is usually sufficient to capture important mesoscale spatial features in the open ocean, it does not resolve sub-mesoscale features like fronts, small eddies, and filaments around islands. Additionally, bottom reflectance in coastal waters prevents data recovery closer than 4 and 9 km from shore at these spatial resolutions. Moreover, it is a common practice in coastal studies to remove at least one neighboring pixel around shallow areas to limit the impact of adjacency effects and ensure no contamination from bottom reflectance. Therefore, the closest data recovered with a 4 km spatial resolution is most often centered at least 6 km away from all 30 m isobaths. However, most islands in the ocean are smaller than 2 $km^2$. For instance, the median island area in the





2593 km x 2593 km region analyzed around the Fiji Archipelago is $\sim$0.06 $km^2$ with $\sim 86\%$ of all islands smaller than 2 $km^2$.
Therefore, having the closest pixel 6 km away from shore, and a pixel size that is at least twice the size of $\sim 86\%$ of islands,
limits our ability to accurately quantify their IME (see example around Niue island Fig. A2). With the approach presented here,
we can maximize data recovery close to shore while keeping the nominal resolution of 1 km of the operational MODIS and
VIIRS level-2 (L2) products. Ideally, we would produce this type of multi-satellite composite for the entire Pacific Ocean, but
we had to limit our study area to four case-studies around islands of interest due to computational and data storage capacity
limitations. In each case, the maps were large enough (i.e. > 1200 km x 1200 km area) to capture the full extent of the IME
around the group of islands studied and were limited to a maximum size of 2600 km x 2600 km area.

## 2.2 Island Mass Effect Detection

### 2.2.1 Bathymetry, island, and submerged reef databases

We created masks at one kilometer spatial resolution denoting land (land mask) and areas shallower than 30 m depth (shallow
mask) for the studied areas using the General Bathymetric Chart of the Oceans (GEBCO) database. Since a large number of
islands and reefs are smaller than the spatial resolution of the GEBCO database (i.e. 15 arc-seconds corresponding to 463 m
at the equator), we utilized the 30 m spatial resolution global island database (Sayre et al., 2019, 2020) to refine the land and
shallow masks for the study areas. We then extended the shallow mask by one additional pixel to ensure all shallow pixels are
masked. Subsequently, we merged the global island database and the submerged reef database from Messié et al. (2022) into a
single database. To ensure accuracy, we automatically verified all island centroids to confirm their alignment with a land pixel
on the land mask and to ensure their associated land polygon was not significantly smaller than the reported island area in
the global island database. Similarly, we automatically checked all submerged reef centroids to confirm their alignment with
a shallow mask pixel and to ensure their associated shallow mask polygons were not significantly smaller than the reported
reef area in the Messié et al. (2022) database. We manually corrected any discrepancies that were identified when comparing
to the bathymetry data and saved the corrections for reference. For simplicity, the term "islands" in this study also refers to
submerged seamounts or reefs shallower than 30 m depth.

### 2.2.2 IME contour delineation

The [Chla] contour value delineating the IME was determined in three successive steps to dynamically detect detached IME
patches. The first step used the method from Messié et al. (2022) to detect IMEs on each 8-day composite map of the time-series
(see Fig. 1.a: step 1). This method defines the [Chla] contour value with an iterative process starting from the highest (chl_max)
to the lowest [Chla] (chl_min) values detected one pixel away from the 30 m isobath of each island and ending when a set of
specified conditions were met. These conditions include: (1) when [Chla] values fall below chl_min, (2) when the IME mask
touches the domain borders or a continent masks, and (3) when regions with [Chla] exceeding 80% of the chl_max are detected
farther than 150 km away from the 30 m isobath. This 150 km threshold was set to allow for the detection of water masses that
were detached from an island and advected offshore (denoted as "detached IMEs") but, at the same time, to prevent potential



bias by accounting for non-IME related [Chla] variability far from the island. We observed that this algorithm performed well when the IME is directly adjacent to the 30m isobath of an island and when the IME is spatially homogeneous, with the highest [Chla] values typically located near the island and decreasing with distance from shore (similar to the IME detected on monthly or yearly satellite averages; Messié et al., 2022). Therefore, this method is valuable as the first step for detecting the strongest

200 IME signal that surrounds an island, referred to in this study as $IME_M$ (Fig. 1.a: step 1). However, this approach underestimates the entire extent of an IME when applied on 8-day [Chla] products because it fails to detect elevated [Chla] patches that have been detached from their originating IME or when pixels with $[Chla] > 0.8 * chl\_max$ were detected more than 150 km from the island of origin. Detached IMEs, typically comprised of dynamic filaments and eddies that are quickly advected away from islands, are detectable on 8-day averaged satellite products, but often not captured using monthly or yearly averages such as the

205 products used by Messié et al. (2022). We therefore extended the method proposed by Messié et al. (2022) by adding another set of detection protocols. We utilized modeled daily surface currents (i.e. global ocean ensemble physics reanalysis products provided by Copernicus Marine Services) to predict the general locations of IME patches that detach from islands (Fig. 1 step 2). For clarity, we refer to the detached IME area obtained with this approach as $IME_D$ (Fig. 1 step 3). The sum of both $IME_M$ and $IME_D$ areas (i.e. total IME) is referred to as $IME_T$. The following sequence was applied to detect IMEs in each 8-day

210 median composite of the time-series (Fig. 1):

Step 1: Detection of $IME_M$ (Messié et al., 2022, Fig. 1.a).

Step 2: Prediction of the general location of $IME_D$ by applying the average current u and v vectors from the previous 8-day period (t = -1) to the location of $IME_T$ detected at t = -1 (Fig. 1.d). When step 2 is performed on the first 8-day median of the time-series (t = 0), the surface current at t = 0 is applied to the $IME_M$ detected at t = 0

215     instead (Fig. 1.b).

Step 3: Delineation of $IME_D$ and $IME_T$ using a second round of [Chla] value iteration ranging from the $95^{th}$ to the $5^{th}$ percentiles of [Chla] measured within the predicted zone and only keeping the patches that overlap with the predicted zone location as explained below.




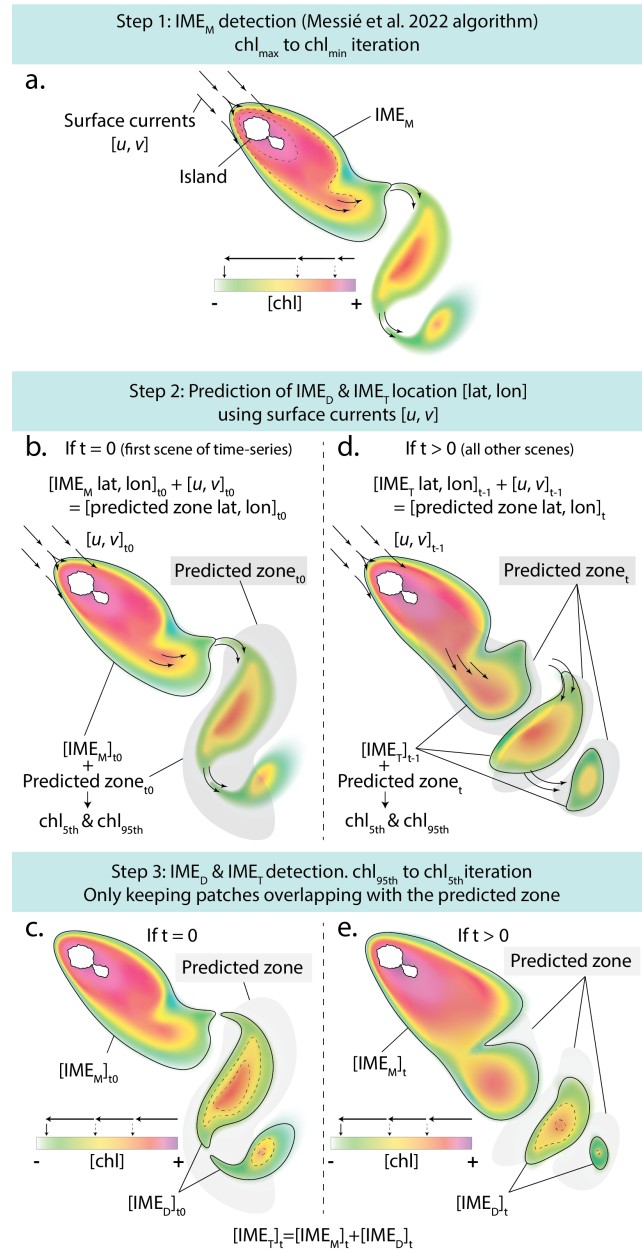

**Figure 1.** Island mass effect detection method. **a.** Step 1: $[IME_M]_{t_0}$ detection following the method from Messié et al. (2022), **b.** step 2 at t = 0 (first image of the time-series): prediction of the detached IME ($[IME_D]_{t_0}$) location applying $t_0$ surface currents ($[u,v]_{t_0}$) to $[IME_M]_{t_0}$ location, **c.** step 3 at t = 0: detached IME contour detection ($[IME_D]_{t_0}$) iterating from the 95th to 5th percentile of [Chla] ($chl_{95th}$ and $chl_{5th}$ respectively) detected within the $[IME_M]_{t_0}$ and the $t_0$ predicted zone, **d.** step 2 at t > 0 (rest of the time-series): prediction of $[IME_D]_t$ location applying $t_{-1}$ surface currents ($[u,v]_{t_{-1}}$) to the total IME location detected on the previous image ($[IME_T]_{t_{-1}}$), **e.** step 3 at t > 0: $[IME_D]_t$ contour detection iterating from $chl_{95th}$ and $chl_{5th}$ detected within the $[IME_T]_{t_{-1}}$ and the t predicted zone.



Step 3 of the detection involves a second round of [Chla] iteration which is based on the $\text{IME}_M$ detection method but
adapted to the higher resolution satellite composites. First, we modified the detection of the [Chla] range, defining the range
of iteration for a given IME, to better capture the dynamic range in [Chla] of the entire IME while avoiding potential biases
in pixels adjacent to the island due to bottom reflectance and adjacency effect. We performed the [Chla] iteration from the
$95^{th}$ to the $5^{th}$ [Chla] percentiles of the entire predicted zone ($\text{chl}_{95th}$ and $\text{chl}_{5th}$) instead of performing the [Chla] iteration
from chl_max to chl_min of the first pixel band around the 30 m isobath of each island. Additionally, the iteration step size was
automatically defined to always correspond to 30 [Chla] steps within the [Chla] range of the entire predicted zone (from $\text{chl}_{95th}$
to $\text{chl}_{5th}$). The number of [Chla] iteration steps (i.e. 30 iterations) was optimized by trial and error to better detect IME around
Rapa Nui, where the [Chla] dynamic range is the lowest and where a small change in [Chla] contour has the most impact
on the IME surface detected. Similarly to the $\text{IME}_M$ detection, once the [Chla] contour value was found, the iteration was
performed again starting at the preceding iteration but with an iteration step size divided by 10 in order to delineate the IME
patch more accurately. As a result, the [Chla] iteration step value ranged from $10^{-4}$ to $10^{-1}$ mg m$^{-3}$ which, in low dynamic
range regions, is smaller than the $10^{-3}$ mg m$^{-3}$ step value used in Messié et al. (2022), and smaller than the accuracy of
absolute [Chla] retrieval from satellites ($10^{-1}$ mg m$^{-3}$, discussed below). This smaller [Chla] iteration step value improved the
performance of the detection algorithm around islands in regions with a very low dynamic range in [Chla] (e.g. Rapa Nui). We
also modified the conditions to stop the [Chla] iteration, removing the condition that stopped the [Chla] iteration when pixels
with $[Chla] > 0.8 * chl\_max$ are located more than 150 km away from the studied island to allow the detection of detached
IME further than 150 km away from the island (i.e. condition number 3; Messié et al., 2022). Additionally, instead of stopping
the [Chla] iteration when the IME touched the domain border, the IME was considered to be exiting the domain and the iteration
was stopped when, for a given [Chla] contour, more than 25% of the predicted pixel location overlapped with a chlorophyll
patch touching the border. This modification improved detection of IME by tolerating a small proportion of the IME patch to
be advected near the domain border while still stopping the iteration when the [Chla] contour becomes too low and includes
features that are not part of the IME. We also added a condition to stop the $\text{IME}_D$ [Chla] iteration when the $\text{IME}_D$ [Chla]
contour intersected an $\text{IME}_D$ contour associated with another island. Finally, as in Messié et al. (2022), the BO reference zones
associated with each IME zone (i.e. $\text{IME}_M$, $\text{IME}_D$, and $\text{IME}_T$) were defined as the area equal to the size of the corresponding
IME zone but located outside of the IME zone, closest to the shallow mask (i.e. BO zone associated with $\text{IME}_M$ denoted as
$\text{BO}_M$ and BO zone associated with $\text{IME}_T$ denoted as $\text{BO}_T$, Table 1). The difference in average [Chla] and $\sum[Chla]_{IME_T}$
between the IME and their corresponding BO reference zone were computed to estimate the biomass increase associated
with an IME relative to the BO (i.e. $\Delta[Chla]_{IME_T - BO_T}$ and $\Delta\sum[Chla]_{IME_T - BO_T}$ respectively). The [Chla] enhancement
attributed to a given IME was deemed significant when both the mean and integrated values were above their uncertainty, e.g.
$\Delta[Chla]_{IME_T - BO_T} - SEM^f_{\Delta[Chla]_{IME_T - BO_T}} > 0$ or $\Delta\sum[Chla]_{IME_T - BO_T} - SEM^f_{\Delta\sum[Chla]_{IME_T - BO_T}} > 0$. Examples of
IME zones detected on the six-month long map time-series around Fiji/Tonga and Samoa/Niue (Fig. 2 and Fig. 3) show
contours outlining the $\text{IME}_M$ (i.e. red contours), the extension of the algorithm to detect the $\text{IME}_D$ (i.e. green contours), and
their associated $\text{BO}_T$ zones (i.e. blue contours). The same analysis was performed around Rapa Nui and the Society Islands,
and are accessible at Bourdin (2024a).



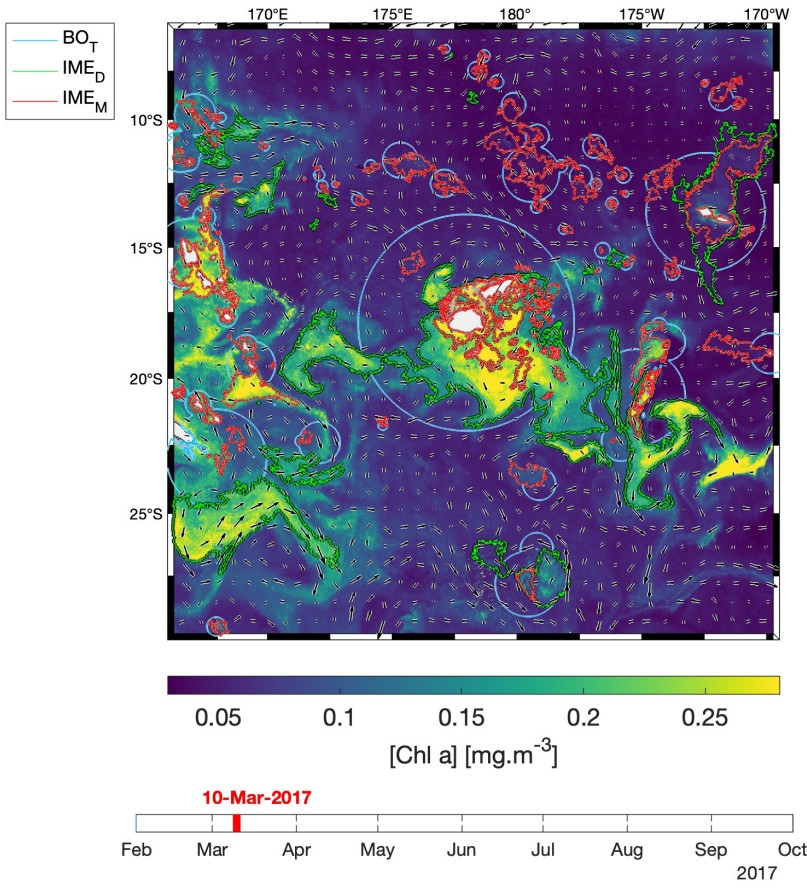

**Figure 2.** Snapshot of six-month long time series of 8-day multi-satellite composites of total chlorophyll *a* concentration ([Chla]) around Fiji and Tonga archipelagos. The $IME_M$ (Messié et al., 2022) contours are delineated in red, the $IME_D$ contours added in this study are delineated in green, and the $BO_T$ associated with each $IME_T$ area is delineated in light blue. Overlaid arrows represent modeled surface current. Entire six-month animated time series accessible in video supplements or at Bourdin (2024a).

### 2.2.3  Detecting IME around neighboring islands

In the case of neighboring islands, it is important to define which island, among a group of islands within a common $IME_M$ patch, contributes the most to the $IME_M$ (referred to as the "lead island"). In Messié et al. (2022), the lead island was defined as the island with the highest chl_min value detected on the first pixel band adjacent to its shallow mask polygon. In our study the 8-day median composite product maps are more spatially heterogeneous than monthly or yearly averages used in Messié et al. (2022) and therefore chl_min values may not be the best indicator to assign a lead island. Moreover, the first pixel band

adjacent to the shallow mask, from which the chl_min value is extracted, is the most likely to be impacted by adjacency effect and bottom reflectance, leading to potential mis-assignment of the lead island. For example, the six-month map time-series around Fiji shows regions of enhanced [Chla] that have been advected in different directions around the archipelago with the





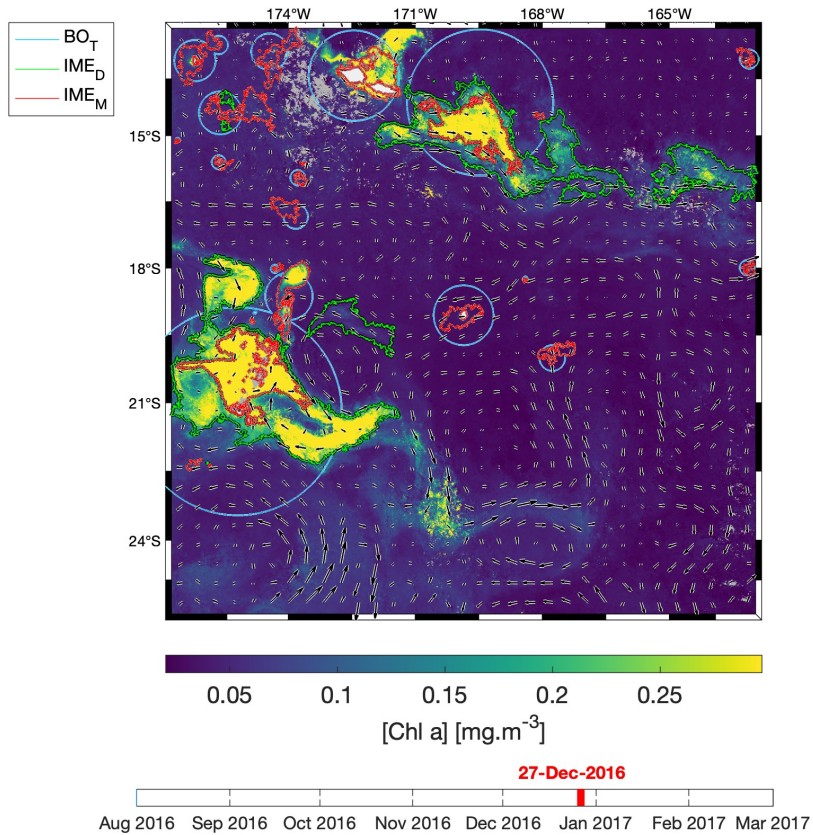

**Figure 3.** Snapshot of six-month long time series of 8-day multi-satellite composites of total chlorophyll *a* concentration ([Chla]) around Samoa (north of the map), Tonga (east of the map), and Niue (center of the map). The $IME_M$ (Messié et al., 2022) contours are delineated in red, the $IME_D$ contours added in this study are delineated in green, and the $BO_T$ associated with each $IME_T$ area is delineated in light blue. Overlaid arrows represent modeled surface current. Entire six-month animated time series accessible in video supplements or at Bourdin (2024a).

largest bloom always centered on Fiji's two largest islands (i.e. Viti Levu = 10912 $km^2$ and Vanua Levu = 5817 $km^2$; Fig. 2). When applying the $IME_M$ criteria, the lead island was assigned to smaller islands (e.g. Koro Island = 105 $km^2$, Yalewa Kalou

Island = 0.2 $km^2$) or to a 20 $km^2$ submerged reef in 19% of the realizations in this time series. Likewise, when applying the $IME_M$ criteria on Society Islands' IME, the lead island was assigned to small islands in 24% of the 8-day frames in the time-series although the bloom was always centered on Tahiti. Based on observations of the time-series of [Chla] maps, we found that for large islands (> 100 $km^2$), the largest IMEs, in terms of area and magnitude [Chla], are generally located around islands with the largest land area. For that reason, in our dynamic model the lead island was reassigned after the $IME_M$ detection (step

1; Fig. 1.a) following a different ranking (see below), which was also later used as the order of detection of the $IME_D$ (Fig. 1 step 3). All islands of a specific study region were first sorted by 100 $km^2$ increments of land area categories (e.g. smaller than 100 $km^2$, between 100 $km^2$ and 200 $km^2$, etc.), then within each category they were further sorted by increments of 10





km$^2$ 30 m isobath area sub-categories (representing the reef area). Thus, land area is ranked higher than reef area only when islands are larger or equal to 100 km$^2$. We further ranked islands within each land area category and reef area sub-category

using their IME intensity based on chl$_{95th}$ values, rounded to the closest 0.1 mg m$^{-3}$. Finally, islands of similar rounded land area, rounded reef area, and rounded chl$_{95th}$ were ranked by their calculated IME$_M$ area. The IME$_T$ detection was performed following this ranking order, thus for a given IME$_T$ zone encompassing multiple islands, the lead island was defined as the top ranked island in the IME$_T$ zone. Once all IME$_T$ detections were performed, the "lead islands" assigned by this ranking were verified to ensure that among all islands associated with a given IME$_T$ patch, the lead island was indeed selected as the first

island in the ranking previously defined. Considering the complexity of the currents around archipelagos, we acknowledge that although a single lead island was assigned to a given IME$_T$, the enhancement in [Chla] associated with IMEs could originate from the influence of multiple islands. For instance, the IME associated with Fiji was a combination of IMEs of all islands and submerged reefs of the archipelago which was also often mixed with the substantial IME influence of the Tonga archipelago. Therefore, IMEs of all islands and reefs associated with archipelagos were combined into "archipelagos IME", such as the

"Fiji-Tonga" IME example (Fig. 2), to track the evolution of the combined IME over the six-month time-series produced (i.e. 88 islands and 140 submerged reefs; Fig. 5). Likewise, the IME$_T$ associated with Samoa encompassed the IMEs of Savaii, Upolu, and Tutuila Islands and all the other small islands and reefs contained within the IME$_T$ patch detected around the archipelago (i.e. 7 islands and 38 submerged reefs; Fig. 4). The IME around Society Islands in French Polynesia were also combined into one large IME that encompassed the Society Islands themselves, the Tuamotu Archipelago, and all small islands

and reefs located in the large IME zone detected around Tahiti (i.e. 176 islands and 34 submerged reefs; Fig. E2). The IME$_T$ associated with Rapa Nui encompassed Rapa Nui and Sala y Gómez islands and two submerged reefs (Fig. E1).

## 3   Assessment

### 3.1   Benefit of multi-sensor composites

Observation and tracking of water masses in the ocean from space is challenging due to glint and clouds that significantly

reduce the amount of data recovered from satellite ocean color sensors. Furthermore, even without clouds or glint, uncertainties associated with satellite retrieval remain substantial mainly due to atmospheric gases (Gilerson et al., 2022). This impact is even larger in oligotrophic and ultra-oligotrophic regions where less light is reflected back to the satellites by the ocean in comparison to the atmosphere. Merging data from multiple satellites with different overpass times and viewing angles offers several advantages: (1) changing cloud coverage over time may allow zones masked by clouds in the morning to be visible

in the afternoon; (2) observing the ocean from varying viewing angles improves data recovery by minimizing the impact of sun-glint; (3) assuming no bias, combining data from sensors with different inherent uncertainties likely reduces the overall uncertainty of the merged product; and (4) as atmospheric properties (other than clouds) change over time, merging data from multiple overpass times can further decrease the relative uncertainty of the final product. Moreover, the correction of adjacency effect and glint by the POLYMER atmospheric correction further increases data recovery and reduces uncertainties around

clouds and in glint impacted areas. By merging products from multiple satellites, we maximized the amount of data available



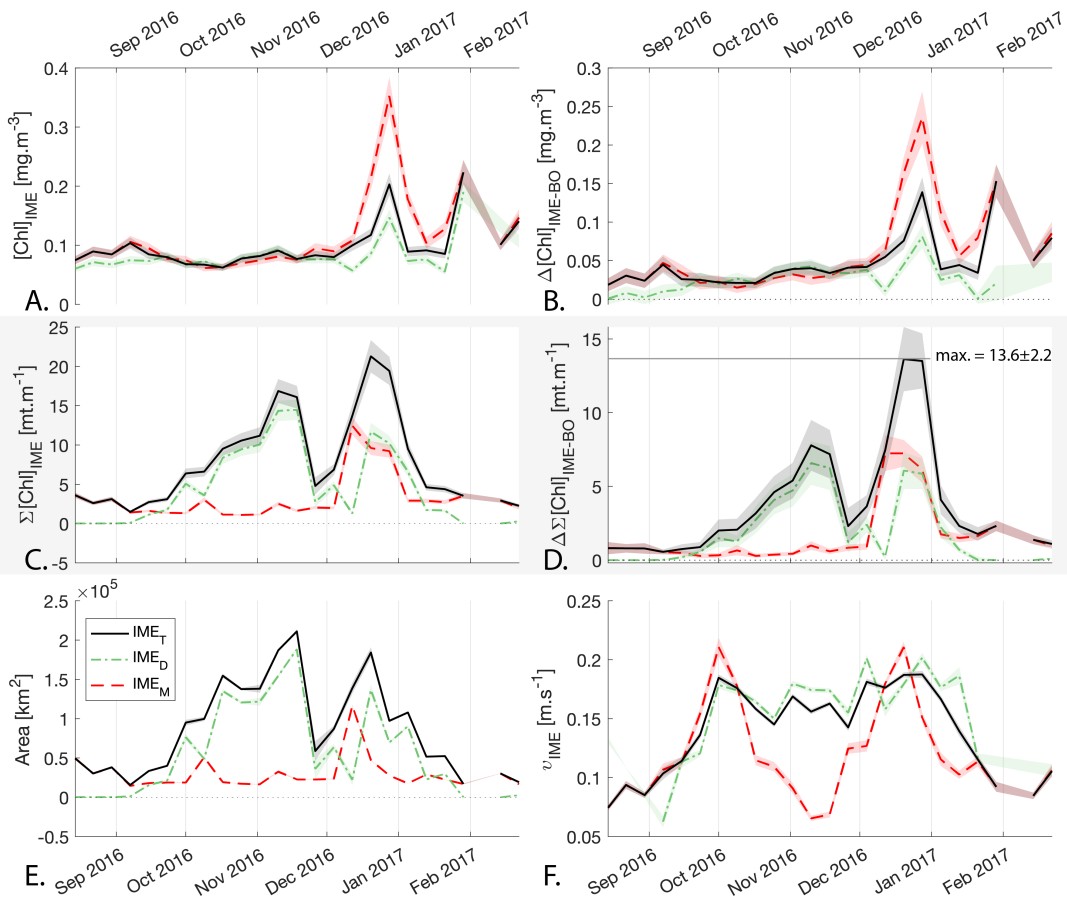

**Figure 4.** Six-month long time series of satellite derived IME properties of the IME zones (IME$_M$ = red dashed line, IME$_D$ = green dash-dotted line, and IME$_T$ = black solid line) detected around Samoa (Savaii, Upolu, and Tutuila). **A and C:** average of properties within the IME zones, **B and D:** difference between properties within each IME zones and their associated BO zones. **A and B:** chlorophyll a concentration ([Chla]), **B and C:** IME integrated chlorophyll a ($\sum[Chla]_{IME}$), **E:** IME zone area, **F:** surface current velocity.

at a given time and location (~10 measurements per pixels in average for a given 8-day period). Recovery of sufficient data for binning was critical to identify and remove outliers, and obtain smooth level-3 products. To further minimize the weight of outliers on the end level-3 products, the binning was performed with medians instead of averages. This method allowed a gap-less and smooth coverage of the zones analyzed during six month time-series at an 8-day frequency, and therefore improve

the detection of sub-mesoscale currents, filaments, and eddies associated with IME.

## 3.2    IME detection algorithm refinement

Time-series of remote sensing maps reveal the complexity of currents around islands and the rather chaotic advection patterns of IME into the open-ocean and between islands (Fig. 2 and Fig. 3). The four case studies were located in the South Pacific





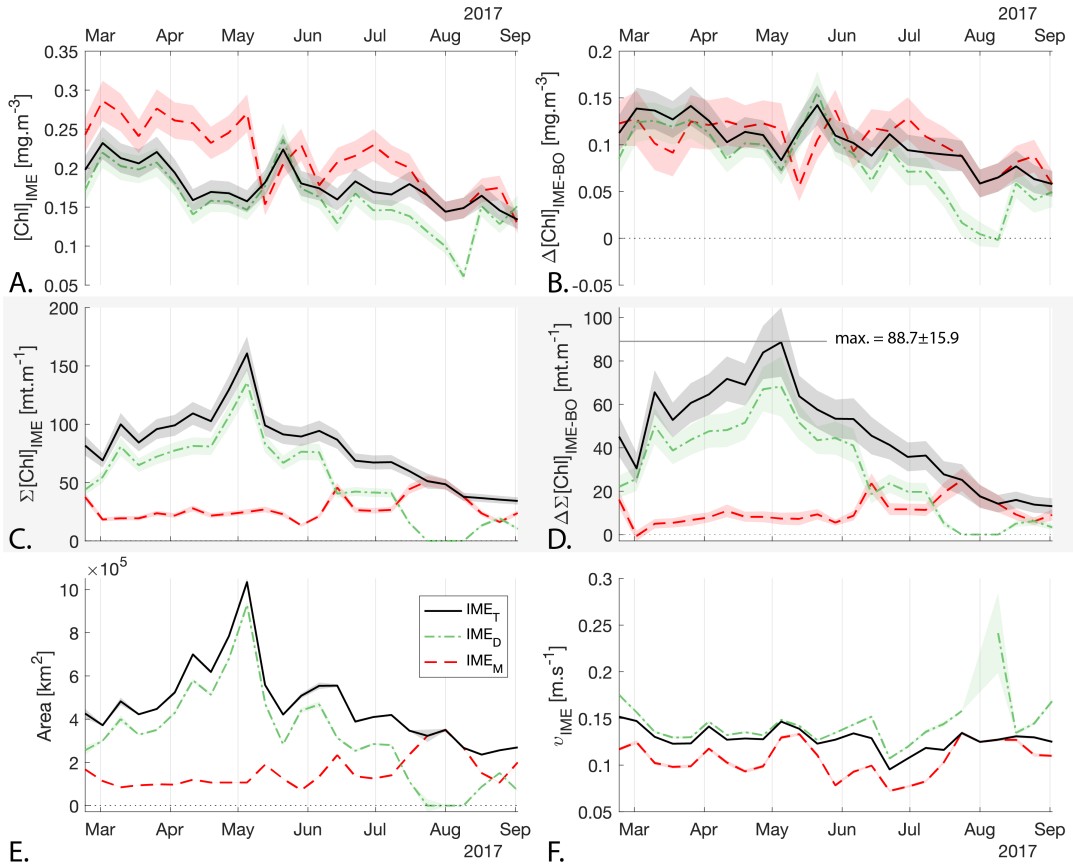

**Figure 5.** Six-month long time series of satellite derived IME properties of the IME zones (IME$_M$ = red dashed line, IME$_D$ = green dash-dotted line, and IME$_T$ = black solid line) detected around Fiji and Tonga archipelagos combined. **A and C:** average of properties within the IME zones, **B and D:** difference between properties within each IME zones and their associated BO zones. **A and B:** chlorophyll a concentration ([Chla]), **B and C:** IME integrated chlorophyll a ($\sum [Chla]_{IME}$), **E:** IME zone area, **F:** surface current velocity.

Subtropical Gyre (SPSG) where geostrophic currents are low and mesoscale and sub-mesoscale currents interact with island

topography from variable directions. In this region, the "upstream" sides of islands also show enhanced [Chla] which suggests IME water masses are advected in all directions around islands (e.g. Fig. A2). Under these conditions and contrary to the assumption in Messié et al. (2022), there are generally no strict upstream pixels directly adjacent to an island. Consequently, defining the lower end of the [Chla] iteration as the minimum [Chla] detected in the first pixel band around the shallow pixel mask may result in an overestimation of the lower threshold of the [Chla] iteration, and thus an underestimation of the IME

area. Therefore, to better capture the local range in [Chla] and to avoid potential remaining impact of adjacency effect and bottom reflectance on satellite retrievals, we extracted the range of the [Chla] iteration from the entire predicted zone of the IME location. In addition, to improve robustness and reduce sensitivity to noise, we used the $95^{th}$ to the $5^{th}$ percentiles instead of the maximum and minimum [Chla] values. By construction, all IME [Chla] were higher than the [Chla] of their




respective BO zones, however, while mean [Chla] of all IME$_T$ zones were significantly higher than their BO$_T$ counterparts
(i.e. $\Delta[Chla]_{IME_T-BO_T} - uncertainty > 0$; Fig. 4, Fig. 5, Fig. E1, Fig. E2), IME$_M$ [Chla] were not significantly higher
than their BO$_M$ counterparts in several occurrences in the eastern SPSG (i.e. $\Delta[Chla]_{IME_M-BO_M} - uncertainty < 0$; Fig.
E1, Fig. E2). This suggests that the larger relative uncertainty in [Chla] retrieval and the very low dynamic range in [Chla] in
this region (Fig. E1) prevented accurate delineation of the entire IME zone using the [Chla] iteration step size of the IME$_M$
algorithm. To improve IME detection in ultra-oligotrophic regions, we used a dynamic [Chla] iteration step size as a function
of the regional [Chla] dynamic range instead of a fixed step size. This adaptive iteration step size resulted in a smaller step size
in ultra-oligotrophic regions than the value used in Messié et al. (2022), and smaller than the accuracy of [Chla] retrieval from
satellites. While a 0.01 mg m$^{-3}$ iteration step is appropriate for accurately delineating IME in mesotrophic regions (Messié
et al., 2022), it represents most of the [Chla] variability of ultra-oligotrophic regions (Fig. A2). Satellite measurements may
exhibit a notable relative uncertainty when retrieving absolute [Chla], particularly in oligotrophic regions. This is mostly due
to the atmospheric contribution being significantly larger than the contribution of the water-leaving radiance to the top-of-
atmosphere radiance measured by satellites (Gilerson et al., 2022). However, given that these Pacific Ocean regions are distant
from major sources of absorbing aerosols, atmospheric properties are expected to be relatively uniform within a specific
satellite image (i.e. MODIS images cover 600 km$^2$ at the equator). Consequently, the precision of the signal necessary to
delineate spatial patterns in [Chla] is expected to be higher than the accuracy of retrieved [Chla]. An advantage of this iterative
method is that it does not rely on absolute values of [Chla] to delineate IME, but rather on spatial increases in [Chla] around
islands. Indeed, reducing the step size of the [Chla] iteration improved the performance of the detection algorithm around small
islands and in ultra-oligotrophic regions where the dynamic range of [Chla] is very low (e.g. Rapa Nui).

In the current study, we adjusted the satellite measurements of [Chla] to best match in situ values and improve our confidence
in accurately retrieving absolute [Chla]. We note that a similar IME delineation accuracy can be achieved, even without in situ
data, by nudging [Chla] of all satellite sensors to one of them to minimize inter-sensor heterogeneity and obtain spatially
homogeneous composites. Even though this method may introduce a bias towards the satellite sensor chosen as reference, this
bias will be equivalent to the bias associated with the use of a single satellite sensor, and, since for the detection of IME we do
not rely on absolute [Chla] values, we expect to achieve a similar accuracy in mapping the extent of IME.

### 3.3 Detached IME detection

When quantifying IME, one challenge is to only account for [Chla] increases associated with this phenomena and not with
other mesoscale processes. Messié et al. (2022) solved this problem by stopping the [Chla] iteration when pixels with $[Chla] >
0.8*chl\_max$ are located more than 150 km away from the 30m isobath of an island. When comparing IME$_T$ and IME$_M$
contours on the same 8-day median [Chla] products, we found that this restriction was the primary reason the IME$_M$ algorithm
underestimated the IME area. With the higher resolution time series obtained here, we show that pixels with the highest [Chla]
within an IME, are heterogeneously distributed and frequently detected further than 150 km from the 30 m isobath. A detection
of such pixel with the IME$_M$ algorithm will result in the termination of the iteration process before the entire IME is detected.
Therefore, in this study, we adapted and improved the IME detection algorithm of Messié et al. (2022) to work with the spatial





and temporal heterogeneity of our level-3 merged satellite products. We removed this aforementioned condition and minimized accounting for potential [Chla] increases due to non-IME related processes by using modelled surface currents to select and

track only the high [Chla] patches that were advected away from islands and submerged reefs. We nonetheless expect a potential overestimation of IME where processes not associated with IME trigger [Chla] accumulation in the surface ocean away from an island and advect this water mass from the open ocean, around an island, and towards the open ocean again downstream of the island (e.g. advection of continental coastal processes, equatorial upwelling, etc.). In these regions, clustering water masses based on more properties than just [Chla] may help differentiate between non-IME [Chla] increases and IME patches. This

clustering method was initially explored in this study using Self-Organizing Maps (SOM; Vesanto and Alhoniemi, 2000) to delineate IME zones based on [Chla], back-scattering coefficient ($b_{bp}$), SST, the ratio of [Chla] and $b_{bp}$, and phytoplankton physiological stress indicators (not shown). While the SOM clustering accurately delineated the IME zones in regions with sufficient dynamic range (e.g. in the western SPSG, around Fiji, or Samoa), the method often failed in the ultra-oligotrophic regions (e.g. in the eastern SPSG around Rapa Nui) where the signal-to-noise ratio of $b_{bp}$ and the physiological stress indices

were too low to delineate IME zones as accurately as the iterative [Chla] method. Therefore, because this study also focuses on regions with relatively low dynamic ranges, we decided not to use the SOM clustering method; nonetheless it could be a good alternative or complement method in regions under continental or upwelling influence where the [Chla] iteration method might overestimate IME. In the four case studies presented here, the high temporal resolution products show that most, if not all, increases in [Chla] initiated close to islands or submerged reefs. The mixed layer depth in the SPSG is almost exclusively

shallower than 80 m, which is significantly shallower than the nutricline in most of the gyre (∼150-220 m; Longhurst, 2007; Raimbault et al., 2008). It implies that wind-driven divergence in this region generally upwells nutrient-deplete water from above the nutricline. In this context, islands and shallow submerged topography may provide the most significant perturbations in this strongly stratified system, with the potential to introduce nutrients to the euphotic zone and trigger phytoplankton blooms as large as the IME zones observed.

### 3.4 IME detection method validation

Consistent with satellite imagery, $IME_M$ and $IME_D$ zones were characterized by elevated underway [Chla] and $c_{p660}$ in comparison to the $BO_T$ zones in all four cases studied (Fig. 6, Fig. D1, Fig. D2, and Fig. D3). Both variables collected with the underway system increased steeply on the inbound transect to Fiji (left hand side panel of Fig. 6) and decreased gradually on the outbound transect (right hand side panel of Fig. 6). Southward currents were the dominant surface currents on the western

side of Fiji during the 16-day period overlapping with in situ sampling. The pattern shown along the outbound transect indicates the demise and/or dilution of the bloom as it was advected south of Fiji. The increase in [Chla] and $c_{p660}$ was ubiquitous near shore and was captured by the satellite $IME_T$ detection algorithm. In comparison, the $IME_M$ algorithm detected the strongest [Chla] increase within IMEs (Fig. 6, Fig. D1, Fig. D2, Fig. D3) but often missed the [Chla] gradient from IME to background ocean (e.g. outbound transect from Society Islands Fig. D2), and systematically missed the $IME_D$ (e.g. inbound transect to

Samoa Fig. D3 and departure from Fiji Fig. 6).

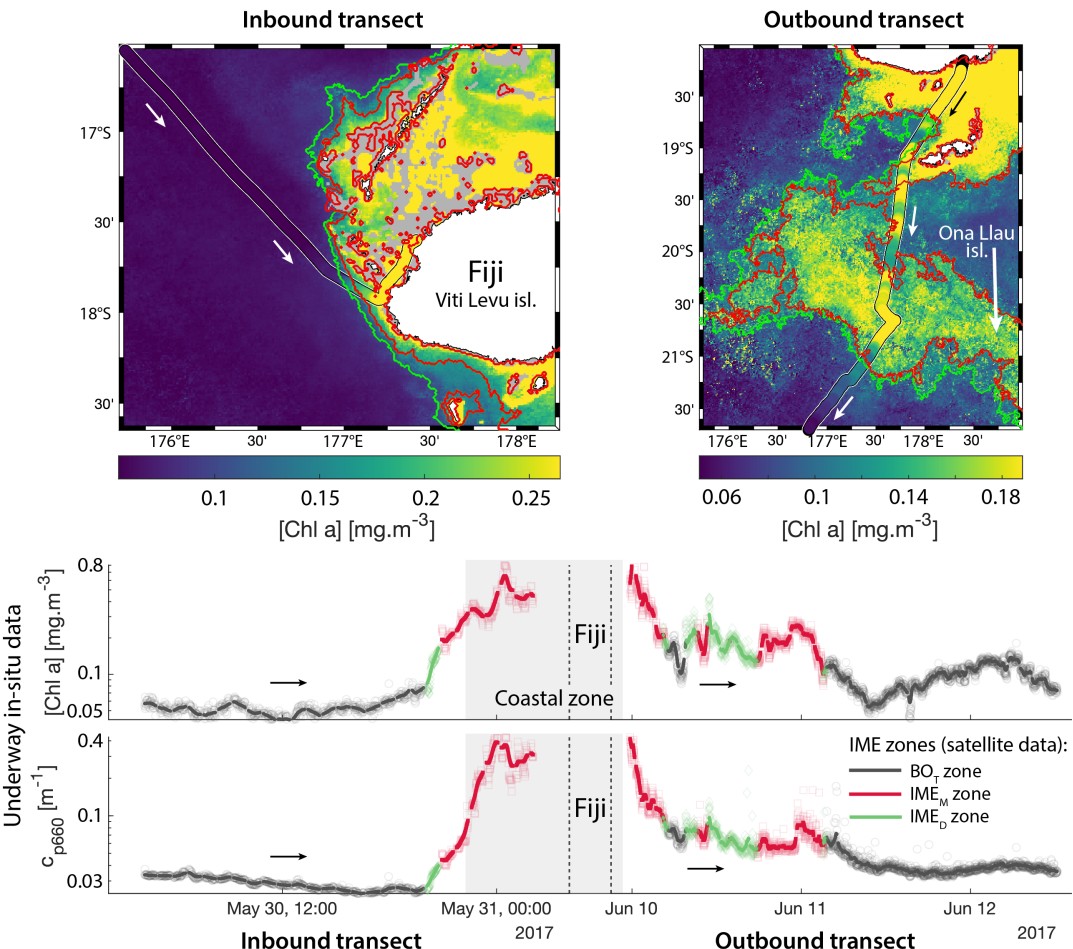

**Figure 6.** Validation of the extent of IME using in situ underway data around Fiji archipelago. **Top row**: 8-day median [Chla] at the time of sampling along the transect inbound to Fiji (top left panel) and at the time of sampling along the transect outbound from Fiji (top right panel). [Chla] measured in situ with the underway system are overlaid on the satellite data background, **middle row**: Chlorophyll a concentration ([Chla]), **bottom row**: beam attenuation at 660 nm (proxy for particulate organic carbon). Data sampled with the underway system during the transect sailing towards Fiji (**left**) and sailing away from Fiji (**right**). Data colored when located within the IME zones detected on the overlapping 8-day satellite composite ($BO_T$ = black circle, $IME_M$ = red square, or $IME_D$ = green diamond). The underway data points are minute binned and the solid lines are smoothed underway data. The smoothing was performed applying a 2h low-pass digital filter to the minute binned data. The grey patch highlights the time *Tara* was sailing in coastal water (< 6 nautical miles away from a submerged reef or coast).

## 3.5 Extent of IME using different algorithms

Similarly, the IME zones detected during the six-month time-series around Fiji/Tonga, Samoa/Niue, Rapa Nui, and the Society Islands (Fig. 2, Fig. 3, and Bourdin, 2024a) suggest that the $IME_M$ detection algorithm generally performs well in capturing





the core of an IME as long as the associated [Chla] distribution is concentric on the island with the highest [Chla] located
close to shore. In all four case studies, the $IME_M$ algorithm generally failed to capture the full extent of the IME area at
8-day observation frequency (i.e. $IME_M$ area $\ll$ $IME_T$ area; Fig. 7 and Table 2). To compare the $IME_M$ algorithm to the one
developed here, we calculated the absolute and percent differences in mean [Chla], detected IME area, and surface-integrated
chlorophyll $a$ ($\sum[Chla]$) derived from the two approaches applied on the same 8-day median [Chla] products (Fig. 7 and
Table 2). [Chla] averages in $IME_M$ zones were equivalent or higher than in the $IME_T$ zones (Fig. 7 and Table 2) because the
minimum value of the [Chla] used in the iteration to find the $IME_T$ contour was always lower than the minimum value used in
the $IME_M$ algorithm. Therefore, when different from the $IME_T$ contour, the $IME_M$ contour was always located closer to the
island shore where [Chla] is generally higher than in the rest of the $IME_T$ zone, explaining the negative differences in average
[Chla] between $IME_T$ and $IME_M$ (Table 2, Fig 7). The area and surface-integrated chlorophyll $a$ were largely underestimated
in $IME_M$ in comparison to $IME_T$ in all four case studies (Fig. 7 and Table 2). For instance, the large bloom event that developed
around Fiji between March and May 2017 detected in the $IME_T$ zone was not detectable in the $IME_M$ zone. The $IME_T$ also
captured a nearly continuous increase in biomass around the Society Islands while it was only intermittently captured by the
$IME_M$ contour (Fig. 7). In each case, the underestimation of $IME_M$ compared to $IME_T$ was variable over time, suggesting the
criteria used to delineate the extent of the $IME_M$ are sensitive to noise in a given satellite image and thus depends on the spatial
smoothness of the [Chla] map used to delineate the $IME_M$. The modification of these criteria in the $IME_T$ algorithm reduced
its sensitivity to single pixel variability.

**Table 2.** $IME_M$ and $IME_T$ detection methods comparison summary: six month mean and standard deviation of differences

| Variables | Island group | $\Delta$ [$IME_T$ - $IME_M$] | $\Delta$[%] |
|---|---|---|---|
| $[Chla]_{IME}$ | Rapa Nui | $7 \times 10^{-4} \pm 5.7 \times 10^{-3} mg.m^{-3}$ | $1\pm10\%$ |
| | Society Isl. | $-2.2 \times 10^{-3} \pm 3.2 \times 10^{-3} mg.m^{-3}$ | $-4\pm6\%$ |
| | Samoa | $-1.7 \times 10^{-2} \pm 3.9 \times 10^{-2} mg.m^{-3}$ | $-14\pm30\%$ |
| | Fiji&Tonga | $-3.6 \times 10^{-2} \pm 3.6 \times 10^{-2} mg.m^{-3}$ | $-21\pm21\%$ |
| IME area | Rapa Nui | $7 \times 10^{4} \pm 6.3 \times 10^{4} km^2$ | $58 \pm 39\%$ |
| | Society Isl. | $2.2 \times 10^{5} \pm 2.2 \times 10^{5} km^2$ | $33\pm32\%$ |
| | Samoa | $5.7 \times 10^{4} \pm 5.8 \times 10^{4} km^2$ | $49\pm35\%$ |
| | Fiji&Tonga | $3.1 \times 10^{5} \pm 2.2 \times 10^{5} km^2$ | $60\pm28\%$ |
| $\sum[Chla]_{IME}$ | Rapa Nui | $4.1 \pm 4.1 mt.m^{-1}$ | $58 \pm 27\%$ |
| | Society Isl. | $10.4 \pm 11.7 mt.m^{-1}$ | $32\pm31\%$ |
| | Samoa | $4.6 \pm 4.9 mt.m^{-1}$ | $45\pm34\%$ |
| | Fiji&Tonga | $52.3 \pm 35.2 mt.m^{-1}$ | $58\pm27\%$ |



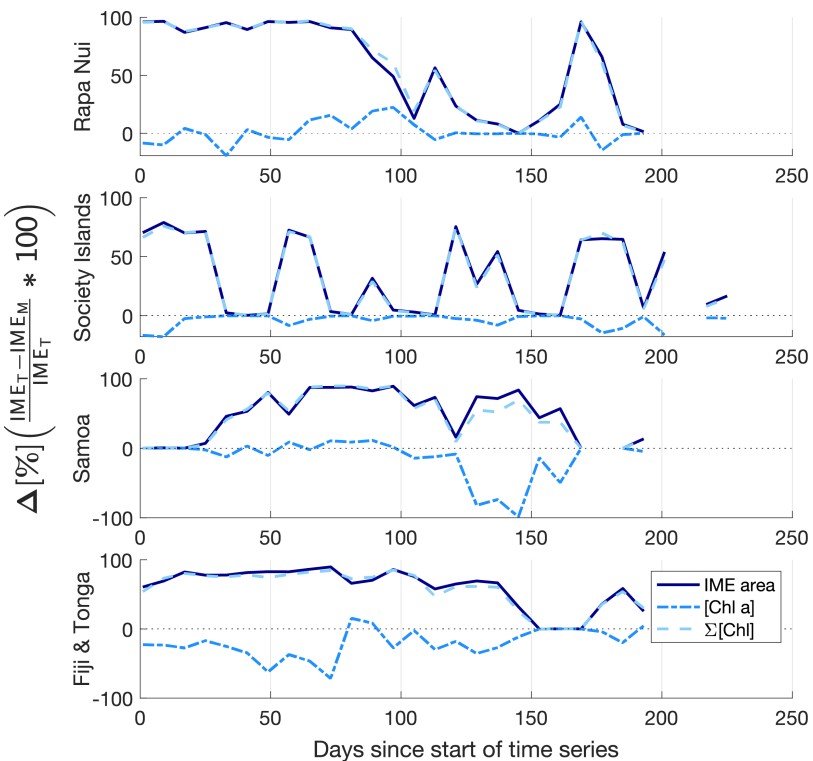

**Figure 7.** Differences (%) in IME area (solid line), chlorophyll a concentration ([Chla]; dash-dotted line), and IME surface-integrated chlorophyll $a$ ($\sum[Chla]_{IME}$; dash line) estimated by the $IME_M$ and $IME_T$ algorithms for the four case studies (Rapa Nui, Society Isl., Samoa, Fiji-Tonga)

.

### 3.6 IME quantification metric

The [Chla] enhancement associated with IME was quantified as the difference between surface-integrated [Chla] in a given IME zone and surface-integrated [Chla] in the respective BO zone (chosen to have the same surface area, see Methods) to better represent the total Chla enhancement. In all four cases, the surface-integrated [Chla] enhancement associated with $IME_T$ relative to

their $BO_T$ counterparts was significant during the entire six-month time-series (i.e. $\Delta \sum[Chla]_{IME_T-BO_T} - uncertainty > 0$) except for two 8-day occurrences around Rapa Nui.

     It should be emphasized that [Chla] can be associated with large uncertainties as a measure of phytoplankton biomass due to photo-acclimation, a process of intra-cellular pigment adjustment in response to changes in light and nutrient conditions (Cullen, 1982; Geider et al., 1998). This is especially the case in regions with increased mesoscale activity and upwelling such

as those adjacent to islands. When low-light adapted cells with larger intra-cellular [Chla] are upwelled to the surface, satellites can measure an apparent increase in [Chla] that is not necessarily associated with an increase in biomass (Hasegawa et al., 2008). In all case studies presented here, the increased [Chla] detected in IME zones was associated with increased $c_{p660}$, which





is a proxy of total organic biomass (including phytoplankton biomass) that is not impacted by photo-acclimation (Behrenfeld and Boss, 2006). This observation provides confidence that detected IME zones were indeed associated with spatial increases

in phytoplankton biomass around islands. When investigating the ecological consequences of IME, it is important to note that both satellite data and our underway measurements only describe surface ocean properties and do not inform about the vertical distribution of biomass in IME zones. Gove et al. (2016) showed that the increase in [Chla] associated with IME propagated below the surface and suggested this increase in [Chla] represented a strong increase in biomass at depth. Although strong subsurface chlorophyll maximums (SCM) are generally measured in subtropical regions, most of the SCM signal is often due

to photo-acclimation to low light availability at depth and only associated with a moderate increase in biomass (Kitchen and Zaneveld, 1990; Fennel and Boss, 2003; Furuya, 1990).

## 3.7    The utility of capturing IME's temporal dynamics

The high temporal resolution products revealed the high spatial and temporal heterogeneity of IME and frequent connectivity between IME zones of distant islands. This dynamic IME detection method permitted tracking in time the accumulation of

chlorophyll $a$ in surface waters, which suggested frequent temporal increases in phytoplankton biomass in addition to the spatial increase in phytoplankton biomass already detected around islands. For instance, the accumulation of integrated [Chla] in IME zones suggests the occurrence of two distinct blooms in Samoa's IME zone and a large bloom in Fiji-Tonga's IME zone. These blooms were sustained for weeks while being advected off-shore and eventually detached from the island they originated from (Fig. 2 and Fig. 3). The first one around Samoa was initiated around mid-September 2016 and was advected

southward towards Niue (see area and $\Delta[Chla]_{IME_T-BO_T}$ increases; Fig. 4). The integrated [Chla] of this bloom continued to increase after the water mass detached from Samoa and persisted near Niue until the end of November 2016 (i.e. $\sim$10 weeks after detaching from Samoa; Fig. 3 and Fig. 4). The second bloom detected in Samoa's IME initiated around November $22^{nd}$ 2016 was advected east, detaching from the archipelago and reaching a maximum surface-integrated [Chla] enhancement relative to BO of 13.6 $mt.m^{-1}$ before ending around January $24^{th}$ 2017 (Fig. 3, Fig. 4). A third bloom observed in the

same region detached from Tonga and was detected more than $\sim$1300 km east of the island. Phytoplankton biomass can continue to accumulate in advected water masses even without an additional influx of nutrients. For example, if the rate of horizontal dilution of a bloom with its surrounding oligotrophic waters reduces encounter rates, and hence grazing pressure, phytoplankton biomass will continue to accumulate even if the remaining nutrients only support a low growth rate (as long as the growth rate exceeds the grazing rate; Lehahn et al., 2017). Interestingly, both bloom initiation events detected around

Samoa were synchronised with a sudden increase in the average surface current velocity within the $IME_M$ zone. The increased current interacting with the island topography may have promoted sub-mesoscale and mesoscale mixing and the upwelling of nutrient and trace metal enriched water to the surface close to shore. The current data overlaid on the [Chla] map time-series also show increased surface current close to shore when and where each of the three blooms started to detach from their island of origin (Bourdin, 2024a). This suggests that when IME water parcels were detached from their source of nutrients (i.e. the

island) and diluted into the surrounding oligotrophic ocean, the phytoplankton biomass in the growing patch continued to accumulate due to a reduction of grazing while using the limited nutrient supply advected with it. This dynamic emphasizes



the fact that although phytoplankton blooms in IME zones are triggered by local enrichment of macro-nutrients and trace metals near islands (Messié et al., 2020, 2022; Gove et al., 2016, 2013; De Verneil et al., 2017; Hasegawa et al., 2009; Caputi et al., 2019; Palacios, 2002; Signorini et al., 1999), they are also tightly controlled by loss processes such as grazing. In the

case of Fiji-Tonga, the IME surface-integrated [Chla] enhancement relative to the BO (i.e. $\Delta \sum [Chla]_{IME_T - BO_T}$) increased up to 88.7±15.8 $mt_{[Chla]}.m^{-1}$, covered an area up to ∼1 million km$^2$, with a longitudinal extent of ∼2000 km. The IME surface-integrated [Chla] decreased from May $5^{th}$ to September $2^{nd}$ to finally reach pre-bloom values again in August 2017, approximately five months after the bloom initiated; Fig. 5). In contrast to the Samoa case study, no apparent increase in current speed was detected near Fiji or Tonga during the period covered by the time series. In this case, the timing of this large

bloom observed around the Fiji and Tonga archipelagos coincided with the annual *Trichodesmium* spp. blooms observed in this region during the austral summer (Dandonneau and Gohin, 1984; Dupouy et al., 2000). The high underwater volcanic activity characteristic of this region can supply significant amount of trace metals directly into the euphotic zone and support these large blooms of *Trichodesmium* spp. (Bonnet et al., 2023; Guieu et al., 2018; Berman-Frank et al., 2001; Lory et al., 2022; Rubin et al., 2011). These known shallow hydrothermal vents were systematically located within the detected IME zone associated

with Tonga and Fiji suggesting the detected IME is likely a combined effect of islands and shallow hydrothermal vents in this region. The longer generation time of *Trichodesmium* spp., which allows surface currents to spread them horizontally, and their ability to partially escape grazing pressure may explain why these blooms can be maintained for five months and cover a significant area of ∼1 million km$^2$ (Capone et al., 1997; Messié et al., 2020). These two case studies show how the dynamic detection of IME provides information about IME phenology and about island connectivity in comparison to a frozen field

observation of the ocean for which all maps are independent of each other.

## 4 Conclusions

The method developed here describes the history of a given IME with finer resolution, and highlights dynamics that are not detectable using monthly and yearly average remote sensing products. Such a method is essential for improving our mechanistic understanding of IME (e.g. whether the cause is island runoff or upwelling) and the ecological succession within IMEs.

De Falco et al. (2022) highlight the uniqueness of interactions between a given island topography and surrounding wind and current flows, suggesting that phytoplankton responses depend on these interactions. Here we show that IMEs are highly dynamic, they can induce large coherent blooms that can sustain for month while being advected more than 1000 km away from their source. These advected IMEs seed the oligotrophic ocean and other islands with water masses characterized by higher phytoplankton abundance and potentially different species composition than the surrounding oligotrophic ocean such as the

*Trichodesmium* blooms in the south-west Pacific ocean. This analysis reveals a broader spatial extent of IMEs in subtropical regions, suggesting that islands have a greater impact on food web dynamics and biogeochemical processes in these areas, which are traditionally considered oligotrophic. This detection method can also be adapted to track water masses with specific optical properties being advected in upwelling regions or in river plumes. We suggest that future studies use more satellite



variables than just [Chla] in regions where processes other than the one studied can cause elevated surface [Chla] to better
discriminate the underlying processes.

We demonstrated the importance of using gap-less high temporal and spatial resolution satellite products and modeled
surface currents to identify and track sub-mesoscale filaments and eddies associated with IME around islands in the subtropical
Pacific Ocean. We minimized satellite uncertainties by augmenting the number of observations and maximized data recovery
by using all available NASA and ESA polar-orbiting ocean color satellites. At the current dawn of global hyperspectral ocean
color sensing, we recommend having sensors with different overpass times when planning for new ocean color satellites as part
of the future constellation to help maximize coverage and understand the dynamic of mesoscale and sub-mesoscale processes
in the Ocean.

*Code and data availability.*    HPLC data is accessible on BCO-DMO repository. In situ underway optical data can be accessed on Tara Pacific
SeaBASS repository. The satellite binning software package used to create custom level-3 multi-satellite products from level-2 satellite
data, to remove outliers, to nudge, and propagate uncertainties is accessible at Bourdin (2024b). Level-3 multi-satellite composites data,
downloaded current data, the dynamic IME detection algorithm software, and its main outputs for each case study, including island databases
for all region and their IME and BO masks, are available at Bourdin (2024a).

*Author contributions.*    TPC supported the collection of in situ data and logistics and G.G., F.L., and E.B. designed and coordinated in situ
sampling. G.B. collected and processed underway in situ data. G.B. designed the satellite merging method and the dynamic IME detection
method. G.B., L.K.B., and E.B. assessed the method and wrote the original draft. All authors have read and reviewed the manuscript.

*Competing interests.*    The authors declare no competing interests

*Acknowledgements.*    Special thanks to the Tara Ocean Foundation, the R/V Tara crew and the Tara Pacific Expedition Participants (Tara
Pacific Consortium). We are keen to thank the commitment of the following institutions for their financial and scientific support that made
this unique Tara Pacific Expedition possible: CNRS, PSL, CSM, EPHE, Genoscope, CEA, Inserm, Université Côte d'Azur, ANR, agnès
b., UNESCO-IOC, the Veolia Foundation, the Prince Albert II de Monaco Foundation, Région Bretagne, Billerudkorsnas, AmerisourceBer-
gen Company, Lorient Agglomération, Oceans by Disney, L'Oréal, Biotherm, France Collectivités, Fonds Français pour l'Environnement
Mondial (FFEM), Etienne Bourgois, and the Tara Ocean Foundation teams. Tara Pacific would not exist without the continuous support
of the participating institutes. The authors also particularly thank Serge Planes, Denis Allemand, and the Tara Pacific consortium. This
work could not have been completed without the support of grants from the NASA Ocean Biology and Biogeochemistry program (grants
#NNX13AE58G, #NNX15AC08G, and #80NSSC20K1641) and NSF Biological Oceanography program (grant #2025402) to the University
of Maine. We gratefully acknowledge Monique Messié for her insightful discussions on the Island Mass Effect and for generously sharing
her island and submerged reef database. This is publication number 27 of the Tara Pacific Consortium.



## Appendix A: Satellite merging pipeline

MODIS, VIIRS, and OLCI L1A radiance data were processed with SeaDAS l2gen and POLYMER algorithms to produce
atmospherically corrected level-2 $R_{rs}$ data. Low-quality data pixels were removed by applying the recommended atmospheric
correction flags on their respective $R_{rs}$ data. Every scene was then projected onto the same equally spaced one-kilometer spatial
resolution plate-carré reference grid using nearest-neighbor interpolation before [Chla] computation. Each satellite [Chla] were
nudged to best match in situ values before merging them into 8-day median composites (Fig. A1).

**Figure A1.** Satellite composite production flowchart.



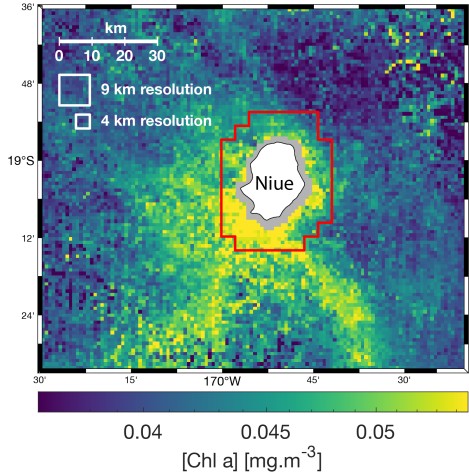

**Figure A2.** Multi-satellite composite of [Chla] around Niue island (2016-09-11 to 2016-09-18) at 1 km spatial resolution. The white squares represent the 4 and 9 km resolution pixel sizes of the level-3 NASA and the red contour represents the shallow pixel mask at 4 km spatial resolution.

## Appendix B: Uncertainty estimates

The 8-day merged products represent a composite of multiple overpasses and satellites that included ∼120 ocean color images (daytime) for a ∼2500 $km^2$ square region around the Fiji archipelago. Therefore, each pixel of the merged product is a median of $n$ number of observations of the original images with standard deviations ($\sigma_{V_{bin}}$) representing the temporal variability of a variable $V$ in a given pixel during each 8-day period and the variability between sensors after nudging. The number of non-flagged observations ($n_{V_{bin}}$) used to bin each merged pixel was generally sufficient, with 8-day long periods and an operational
constellation of five to six satellites, to produce smooth merged [Chla] products. For example, the median number of non-flagged [Chla] observations used to bin each pixel was $n_{bin_{[Chla]}} = 10$ for the entire time series around Fiji, with less than 2.5% of the pixels binned with less than 3 non-flagged observations (Fig. B1).

Known uncertainties were propagated from in situ data to satellite [Chla] end-products. HPLC derived [Chla] and in situ $a_p$ spectra were measured along track. The error associated with the computation of [Chla] from the underway system was
estimated by the normalized root mean square error ($nRMSE_{udw}$ in %) of the relation between the underway chlorophyll line height ($a_{p676LH}$) and total [Chla] measured from HPLC during the *Tara* Pacific expedition (Fig. B2.a).

The error associated with the computation of [Chla] from satellites was estimated by the $nRMSE_{sat}$ of the relation between the underway chlorophyll line height ($a_{p676LH}$) and [Chla] obtained from each satellite sensor along the transect of the *Tara* Pacific expedition (Fig. B2 (b), (c), and (d)). The uncertainties of binned satellite end-products were computed as follows:

$$\sigma_V = \sqrt{\sigma_{Vbin}^2 + \sum_{n=1}^{n_c} (\tilde{V} \times nRMSE_c)^2} \tag{B1}$$





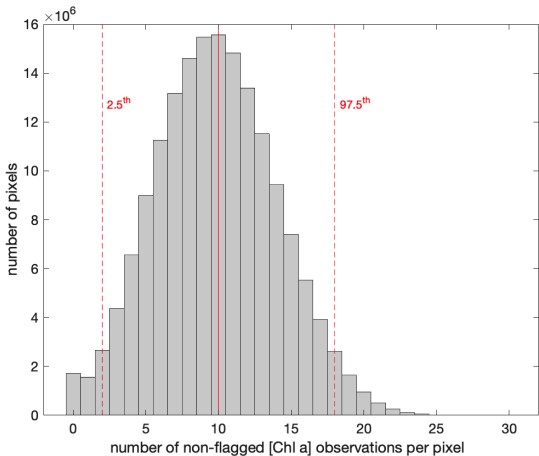

**Figure B1.** Distribution of the number of valid [Chla] (i.e. not flagged) observations per merged pixel over each 8-day period along the six-month time-series around the Fiji archipelago (2017-02-18 to 2017-09-05).

**Table B1.** Robust correlations parameters of match-ups between satellite and in situ underway data

| Variables | Satellite sensor | $R^2$ | nRMSE [%] | Slope | Intercept | N |
|---|---|---|---|---|---|---|
| POLYMER $R_{rs}$ Blended CI-OCx [Chla] (chlor_a) | MODISA | 0.78 | 24.38 | 1.09 | -0.01 | 111 |
| | MODIST | 0.81 | 20.48 | 1.08 | -0.01 | 96 |
| | VIIRSN | 0.82 | 16.18 | 0.90 | -0.19 | 109 |
| | VIIRSJ1 | 0.70 | 31.47 | 1.02 | 0.05 | 27 |
| | OLCI | 0.79 | 16.56 | 0.89 | -0.13 | 85 |
| SeaDAS $R_{rs}$ Blended CI-OCx [Chla] (chlor_a) | MODISA | 0.74 | 18.59 | 0.84 | -0.27 | 85 |
| | MODIST | 0.71 | 16.61 | 0.70 | -0.43 | 67 |
| | VIIRSN | 0.70 | 16.17 | 0.81 | -0.56 | 92 |
| | VIIRSJ1 | 0.84 | 17.37 | 1.15 | 0.14 | 22 |
| | OLCI | 0.81 | 13.99 | 1.01 | -0.10 | 55 |
| SeaDAS $R_{rs}$ OCx [Chla] (chl_ocx) | MODISA | 0.66 | 20.00 | 0.71 | -0.32 | 85 |
| | MODIST | 0.67 | 19.30 | 0.75 | -0.35 | 67 |
| | VIIRSN | 0.62 | 25.97 | 1.20 | -0.50 | 92 |
| | VIIRSJ1 | 0.74 | 21.76 | 1.36 | 0.19 | 22 |
| | OLCI | 0.70 | 17.65 | 0.80 | -0.11 | 55 |

With $\tilde{V}$ the binned variable, $n_c$ the number of calibration/correction, and $nRMSE_c$ the $nRMSE$ associated with each of the $n_c$ correction. The standard error of the mean of the adjusted satellite end-products of each pixel were computed as follows:

$$SEM_V = \frac{\sigma_V}{\sqrt{n_{V_{bin}}}} \tag{B2}$$



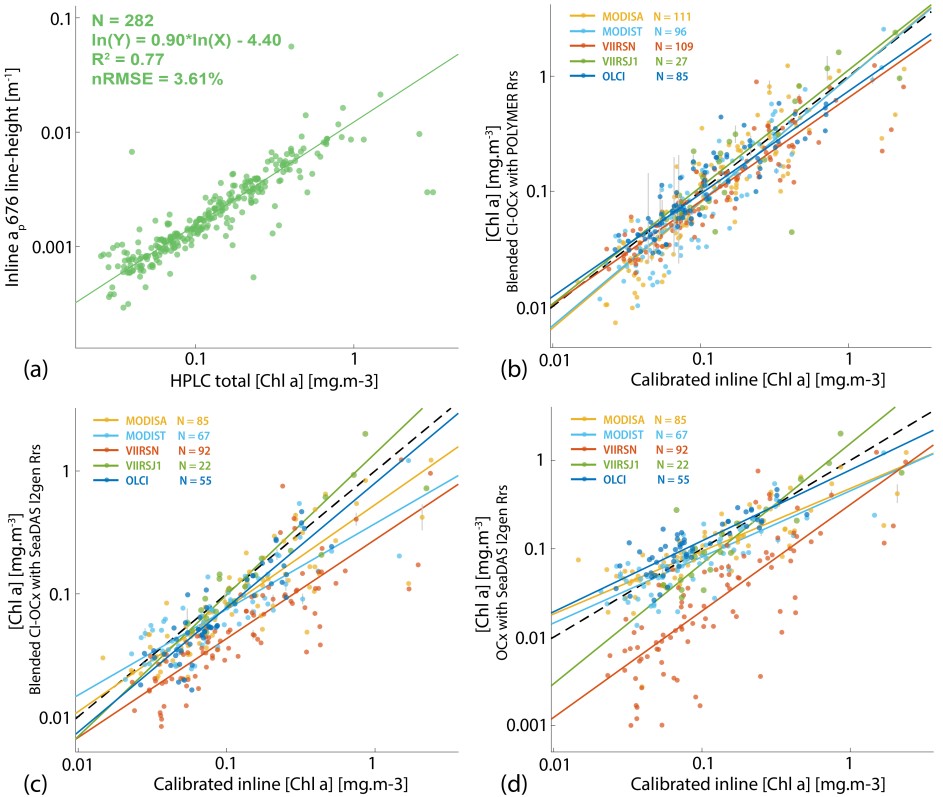

**Figure B2.** Robust linear regressions between [Chla] measured from HPLC and $a_p$ Chla absorption peak at 676 nm measured from the underway system (a), and between calibrated [Chla] estimated from $a_p$ underway measurements and [Chla] estimated from satellite data using the blended CI-OCx algorithm on POLYMER $R_{rs}$ (b), the blended CI-OCx algorithm on l2gen $R_{rs}$ (c), and the OCx algorithm on l2gen $R_{rs}$ (d). In situ measurements were conducted during the *Tara* Pacific expedition (May 2016 to October 2018).

The final uncertainty estimate associated with [Chla] within entire IME or BO zones ($SEM^f_{[Chla]_{IME}}$) as presented in Fig. E1,
Fig. E2, Fig. 4, and Fig. 5 were expressed as the average standard error of the mean of the adjusted [Chla] within entire IME or BO zones:

$$SEM^f_{[Chla]_{IME}} = \frac{\overline{\sigma}_{[Chla]_{IME}}}{\sum n_{[Chla]_{bin_{IME}}}} + S_{[Chla]_{unc}} \times \overline{[Chla]}_{IME} \tag{B3}$$

With $\sum n_{[Chla]_{bin_{IME}}}$ the total number of [Chla] observations within the IME zone before merging and $S_{[Chla]_{unc}}$ the weighted bias associated with the computation of the slopes of the regressions between in situ [Chla] and each satellite [Chla] estimates.
$S_{[Chla]_{unc}}$ was computed as follows:

$$S_{[Chla]_{unc}} = |1 - S_{[Chla]sat}| \times \frac{n_{M_{sat}}}{N_{M_{tot}}} \tag{B4}$$

With $S_{[Chla]sat}$ the slope of the relation between in situ [Chla] and [Chla] of a given satellite, $n_{M_{sat}}$ the number of valid match-ups of the same satellite, and $N_{M_{tot}}$ the total number of valid match-ups. $S_{[Chla]_{unc}}$ represents the maximum bias associated



with the computation of the merged satellite [Chla] which we assume to be equivalent to the potential likelihood bias of the merged satellite [Chla]. Assuming enough valid match-ups with each satellite, $S_{[Chla]_{unc}}$ is a conservative estimate of the bias associated with the slopes computation because the merging method forces each satellite [Chla] to agree with in situ data using sensor-specific corrections, which likely reduces the bias of the merged product. IME area uncertainties ($\sigma_{A_{IME}}$) were computed during the detection of the IME [Chla] contours as the difference in IME area between the last two iterations of [Chla] contours:

$$\sigma_{A_{IME}} = A_{IME_{cChl_f}} - A_{IME_{cChl_{f-1}}} \tag{B5}$$

With $A_{IME_{cChl_f}}$ the IME area at the final IME contour value and $A_{IME_{cChl_{f-1}}}$ the IME area at the previous contour value. Therefore, $\sigma_{A_{IME}}$ represents the area detection resolution associated with the size of the step of [Chla] iteration. The uncertainties associated with the estimation of IME surface-integrated [Chla] ($\sum[Chla]_{IME}$) were computed as follows:

$$SEM^f_{\sum[Chla]_{IME}} = \sum[Chla]_{IME} \times \sqrt{\left(\frac{SEM^f_{[Chla]_{IME}}}{[Chla]_{IME}}\right)^2 + \left(\frac{\sigma_{A_{IME}}}{A_{IME}}\right)^2} \tag{B6}$$

**Appendix C: Outliers removal**

Bio-optical variables in the ocean, including [Chla], generally follow a log-normal distribution (Campbell, 1995) with fewer high values forming a heavy-tail in the high end of the dynamic range. After appropriate flagging, low quality data pixels impacted by sun glint, adjacency effect, and bottom reflectance are rare and account for a few pixels scattered on either end of the log-normal distribution and beyond realistic values for a given region (generally $< 1^{st}$ percentile or $\gg 99^{th}$ percentile; Fig. C1). Computing the median of these pixels can result in noisy merged products when they are the only available data over a given 8-day period and at a given location (i.e. pixel). Consequently, to improve consistency of the level-3 merged products, rare outliers of a given variable were removed from all re-projected level-2 images of a given 8-day period and a given region based on the distribution of all individual $x$ measurements (i.e. pixels). First, we grouped all the re-projected level-2 images of a given variable, 8-day period, and region together, and applied a log normal transformation to the data:

$$x_t = \ln(x - \min(x) + 1) \tag{C1}$$

We partitioned $x_t$ into $N$ bins of width $W$ defined using the Freedman-Diaconis rule that is more suited to a heavy-tailed distribution due to its low sensitivity to outliers (Freedman and Diaconis, 1981):

$$W = 2 \times \frac{IQR(x_t)}{\sqrt[3]{n}} \tag{C2}$$

Where $IQR$ is the inter-quartile range and $n$ is the number of observations in the data $x_t$. The minimum number of pixels per bin threshold ($n^b_{min}$) was computed as a rounded fraction of $n$ of a given variable (i.e. horizontal line in Fig. C1):

$$n^b_{min} = \lfloor n \times 10^{-6} \rceil \tag{C3}$$



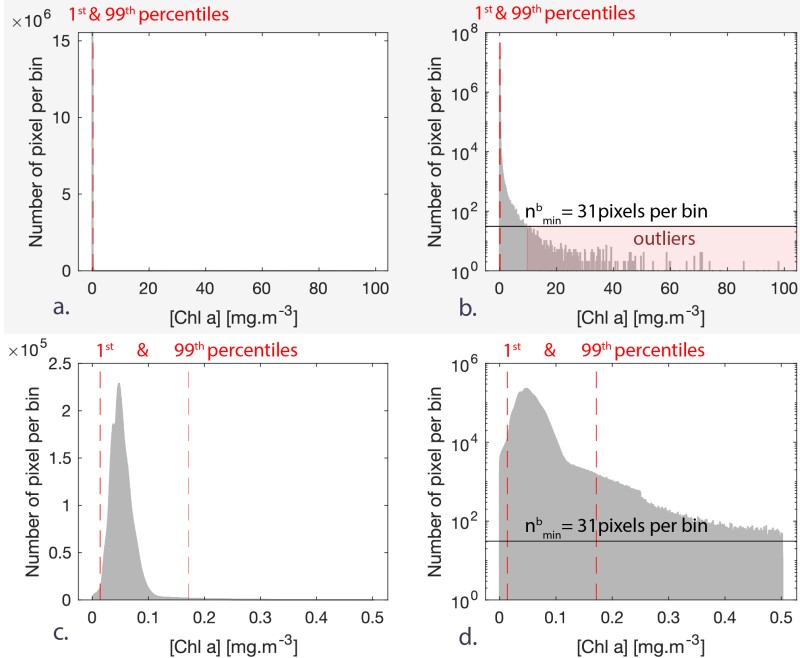

**Figure C1.** Example distribution of all valid [Chla] (i.e. not flagged) observations from all satellite sensors merged (i.e. MODISA-Aqua, MODIS-Terra, VIIRS-SNPP, OLCI-S3A) from 2016-09-19 01:00 to 2016-09-26 21:30 UTC (8-day period) around Niue and Samoa (77 satellite images merged) before outlier removal (subplots (a) and (b)), and after outlier removal (subplots (c) and (d)). The number of pixels per bin are displayed on a linear scale on subplots (a) and (c) and on a log base 10 scale on subplots (b) and (d). The dashed lines represent the $1^{st}$ and the $99^{th}$ percentiles, the solid horizontal line represents the cut-off value in pixel per bin for outlier removal ($n_{min}^{b}$), and the red shaded area highlights the pixels removed.

The lower-end threshold $t_L$ was determined by finding the first bin with less pixels than $n_{min}^b$ (i.e. gap in normal distribution), going from the median $\tilde{x}$ to $x_{min}$ ($x_{min} \leq t_L < \tilde{x}$). Similarly, the higher-end threshold $t_H$ was determined by finding the first bin with less than $n_{min}^b$ pixels per bin, going from $\tilde{x}$ to $x_{max}$ ($\tilde{x} < t_H \leq x_{max}$). This threshold detection was iterated up to 585 15 times or until $t_L$ and $t_H$ did not change from one iteration to the other. Any re-projected level-2 pixel from a given 8-day period, region, and variable falling out of the range ($t_L$, $t_H$) were deleted before computing the medians of the merged level-3 products.



## Appendix D: Rapa Nui, Society Isl., and Samoa validation transects

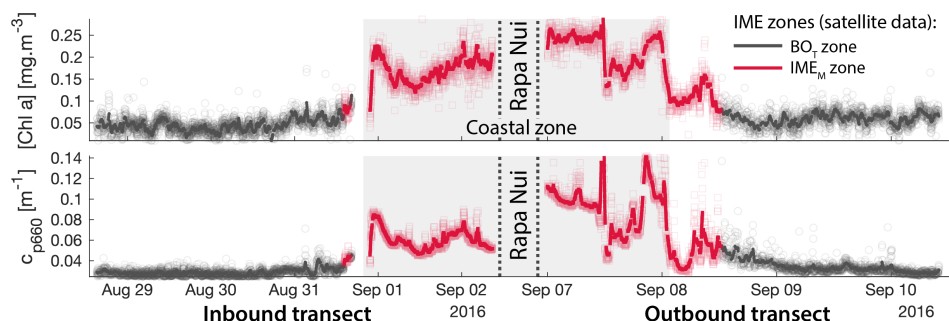

**Figure D1.** Validation of the IME extent using in situ underway data around Rapa Nui. **Top row**: Chlorophyll a concentration ([Chla]), and **bottom row**: beam attenuation at 660 nm (proxy for particulate organic carbon). Data sampled with the underway system during the transect sailing towards Rapa Nui (**left**) and sailing away from Rapa Nui (**right**). Data colored when located within the IME zones detected on the overlapping 8-day satellite composite ($BO_T$ = black circle or $IME_M$ = red square). The points are minute binned underway data and the solid lines are smoothed underway data. The smoothing was performed by applying a 2h low-pass digital filter to the minute binned data. The grey patch highlights the time *Tara* was sailing in coastal water (< 50 m depth).

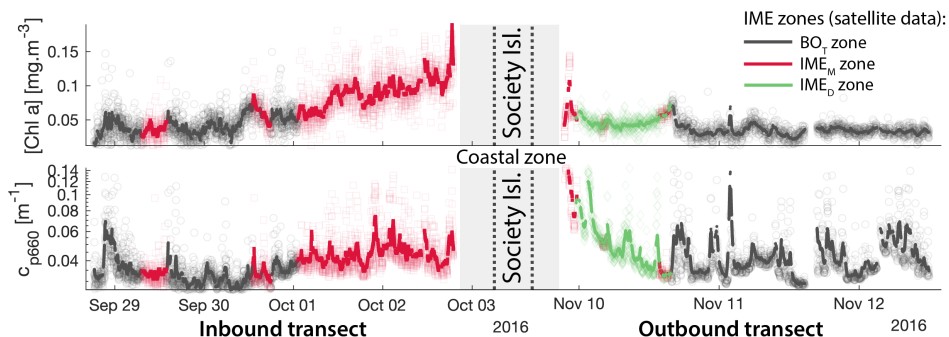

**Figure D2.** Validation of the IME extent using in situ underway data around Society Islands in French Polynesia. **Top row**: Chlorophyll a concentration ([Chla], **bottom row**: beam attenuation at 660 nm (proxy for particulate organic carbon). Data sampled with the underway system during the transect sailing towards Society Islands (**left**) and sailing away from Society Islands (**right**). Data colored when located within the IME zones detected on the overlapping 8-day satellite composite ($BO_T$ = black circle, $IME_M$ = red square, or $IME_D$ = green diamond). The points are minute binned underway data and the solid lines are smoothed underway data. The smoothing was performed applying a 2h low-pass digital filter to the minute binned data. The grey patch highlights the time *Tara* was sailing in coastal water (< 50 m depth).



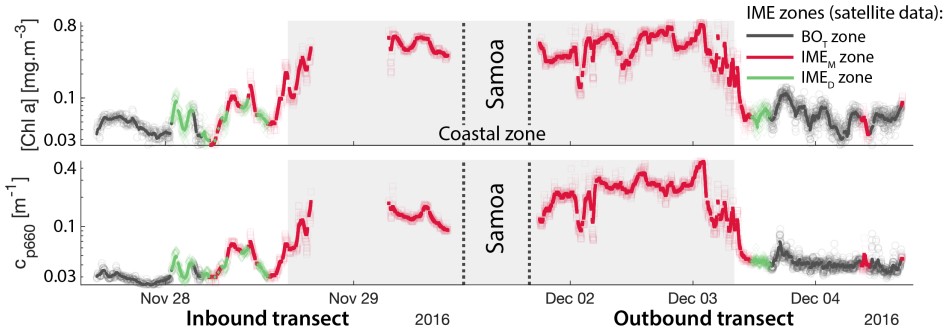

**Figure D3.** Validation of the IME extent using in situ underway data around Samoa. **Top row**: Chlorophyll a concentration ([Chla]), and **bottom row**: beam attenuation at 660 nm (proxy for particulate organic carbon). Data sampled with the underway system during the transect sailing towards Samoa (**left**) and sailing away from Samoa (**right**). Data colored when located within the IME zones detected on the overlapping 8-day satellite composite ($BO_T$ = black circle or $IME_M$ = red square). The points are minute binned underway data and the solid lines are smoothed underway data. The smoothing was performed by applying a 2h low-pass digital filter to the minute binned data. The grey patch highlights the time *Tara* was sailing in coastal water (< 50 m depth).





## Appendix E: Rapa Nui and Society Isl. time-series

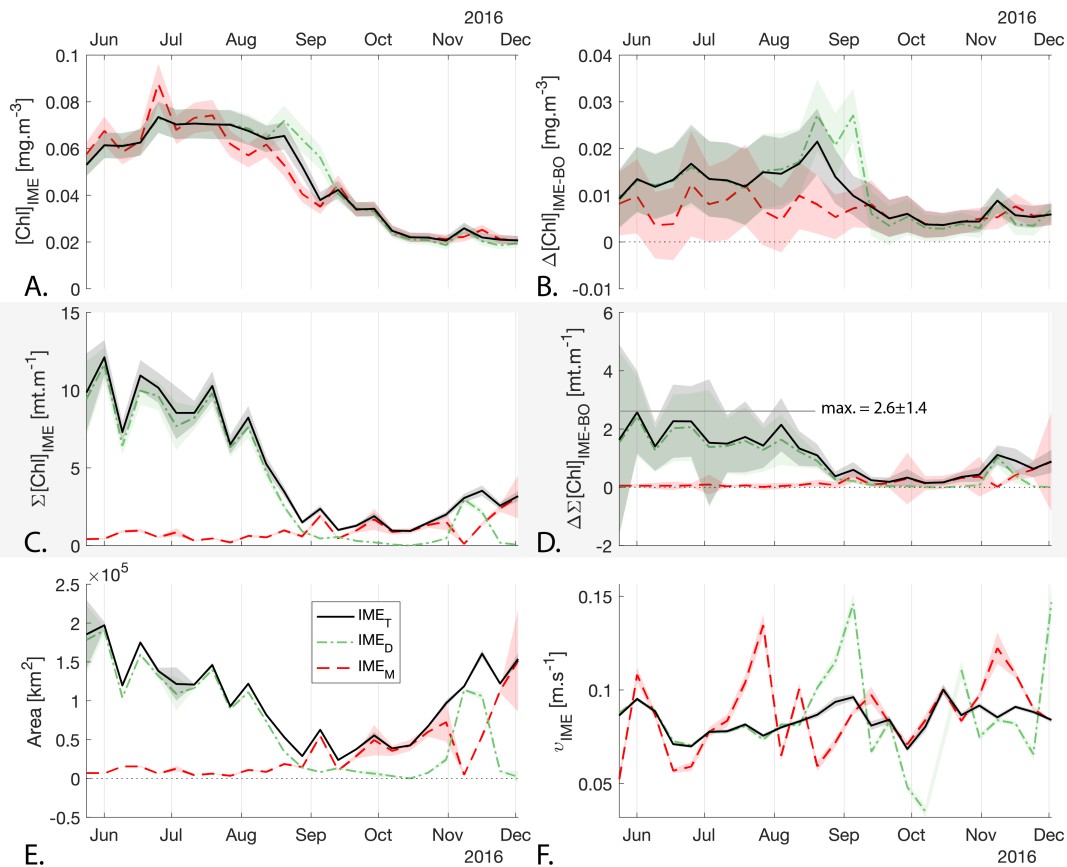

**Figure E1.** Six-month long time series of satellite derived IME properties of the IME zones ($IME_M$ = red dashed line, $IME_D$ = green dash-dotted line, and $IME_T$ = black solid line) detected around Rapa Nui. **A and C:** average of properties within the IME zones, **B and D:** difference between properties within each IME zones and their associated BO zones. **A and B:** chlorophyll a concentration ([Chla]), **B and C:** IME integrated chlorophyll a ($\sum[Chla]_{IME}$), **E:** IME zone area, **F:** surface current velocity.





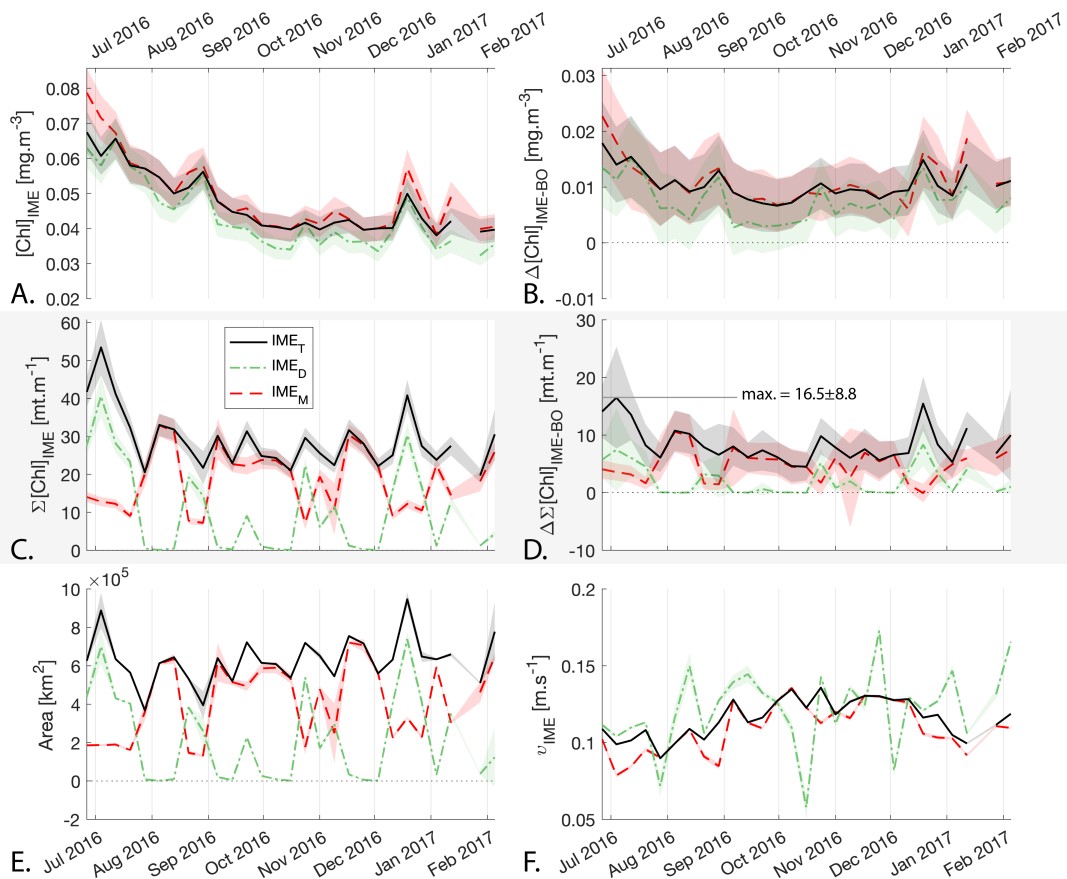

**Figure E2.** Six-month long time series of satellite derived IME properties of the IME zones (IME$_M$ = red dashed line, IME$_D$ = green dash-dotted line, and IME$_T$ = black solid line) detected around Society Islands in French Polynesia. **A and C:** average of properties within the IME zones, **B and D:** difference between properties within each IME zones and their associated BO zones. **A and B:** chlorophyll a concentration ([Chla]), **B and C:** IME integrated chlorophyll a ($\sum[Chla]_{IME}$), **E:** IME zone area, **F:** surface current velocity.




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
