# Peer review of "Dynamics of island mass effect. Part I: detecting the extent"

_EGUsphere, 2024_

## Referee Comment (RC1)

**Dynamics of island mass effect from space. Part I: detecting the extent**

Guillaume Bourdin, Lee Karp-Boss, Fabien Lombard, Gabriel Gorsky, and Emmanuel Boss

**Summary**

The manuscript titled "Dynamics of island mass effect. Part 1: detecting the extent" describes updated algorithms to detect mesoscale and sub-mesoscale processes near remote islands and atolls, termed the "Island Mass Effect" (IME) using satellite remote sensing data. The authors state that existing algorithms for detecting the IME (Messie et al., 2022 underestimate the effect due to using low temporal and spatial resolution satellite data. This study utilizes remote sensing data from multiple sensors to increase temporal resolution and apply a different atmospheric correction scheme (POLYMER) that results in more data. These updated IME algorithms are applied to merged satellite data collected over four island groups in the South Pacific. The results indicate the ecological influence of the IME near these islands is more significant and dynamic than previously thought. The results indicate large phytoplankton blooms that can be advected 1000 km away from their source, seeding the nearby oligotrophic ocean. The overall results of this study indicate that the IME has a greater impact on food web dynamics and biogeochemical processes for waters in close proximity to these remote islands. The authors recommend future studies use higher temporal and spatial resolution satellite products and modeled surface currents to better identify and track sub-mesoscale filaments and eddies associated with the IME around remote islands.

**Major comments**

Introduction
- No major comments

Methods
- The Methods need substantial reorganizing and clarification:

    o More details should be added to the POLYMER atmospheric correction description. Why does it improve data recovery in areas impacted by glint and adjacency effect? What version did you use? Where was it downloaded from? What was it run on? What flags were used? What ancillary data was used?

    o You should also be citing this paper as well: François Steinmetz and Didier Ramon "Sentinel-2 MSI and Sentinel-3 OLCI consistent ocean colour products using POLYMER", Proc. SPIE 10778, Remote Sensing of the Open and Coastal Ocean and Inland Waters, 107780E (30 October 2018); https://doi.org/10.1117/12.2500232

    o The headings in the Methods section seem disorganized to me. For Section 2.1 Level-3 Multi-satellite composites, you start with an intro paragraph and then have several subheadings. Consider merging the intro paragraph into the 2.1.1 section. The 2.1.2 In situ data sub-section seems out of place in this section,

consider adding a new section just for in situ data and matchups. Perhaps this organization with just two sections?

- 2.1 Level-3 satellite products computation
  - (This section will include the writing in the opening paragraph. 2.1.1, 2.1.4, 2.1.5)
- 2.2 In situ data and matchups
  - (This section will include section 2.1.2 and 2.1.3)

- The description of running POLYMER and l2gen should be in same paragraph/section. Right now, you have text on l2gen in the in situ and satellite matchups section which seems out of place.

- In Section 2.1.1, there is no description of how data was processed to Level-3 format. It seems to stop at L2.

- Figure 1 seems okay in the Methods because it is a figure of the workflow. However, should Figures 2-3 be in the "Assessment" section since it is showing the results of the workflow? I don't think you should be referencing results figures in the Methods, save that for the Results (or "Assessment").

Assessment
- Some of the text in the Assessment section would better belong in the Methods such as the description of merging and binning and the chl iteration step size

**Minor comments**

The title on the preprint PDF is different than what is in the system.
Line 5: Consider adding *a* after chlorophyll. Same on Line 20.
Line 8: Define POLYMER
Line 18: The way this sentence is written makes it seem like "their wake downstream.." refers to the winds and currents. Reword to make this more clear.
Line 28: Consider adding the citation to the end of this sentence.
Line 35: Consider changing "They" to "The authors"
Line 77: Should define these satellite mission acronyms
Line 79: More information on the POLYMER atmospheric correction scheme should be included here. See major comment above.
Line 84: Include the time frame of data collected. What is "all"?
Line 84: Why did you download L1A data instead of L1B? Review differences here: https://oceancolor.gsfc.nasa.gov/resources/docs/product-levels/#:~:text=Level%201B%20data%20are%20Level,had%20instrument%2Fradiometric%20calibrations%20applied.&text=Level%202%20data%20consist%20of,the%20source%20Level%201%20data.
Line 85: What Copernicus repository? Provide link(s).

Line 88: What did you use to project the satellite data onto a plate-carre reference grid using NN interpolation?

Line 91: Confused on how this is surface-integrated chla when you're just summing chla concentration in each pixels by the area? Where does depth come into play?

Line 96: Not sure you need to hyphenate hyperspectral

Line 110: Capitalize Python

Line 112: You describe how all satellite data is processed to Level-3 using same scheme as aforementioned but this was never described.. You don't introduce the terms reprojecting, nudging, or merging until now. What is nudging?

Line 114: OCSSW stands for Ocean Color Science Software

Line 119: Consider adding the satellite overpass times for each sensor. How do they match up with the 10:30am local time for in situ data collection?

Line 120: The sentence about recommended Level-2 masks needs a citation. Masks or flags? Did you use recommended L2 or L3 flags? https://oceancolor.gsfc.nasa.gov/resources/atbd/ocl2flags/

Line 120: Are you working with Rrs or nLw? Are these both included when running l2gen and POLYMER?

Line 136: What is GlobColour?

Line 138: Why would this described merging strategy require simulation of 510 nm band?

Line 162: This sentence should have a citation

Line 168: Did you use the 300m spatial resolution of OLCI?

Section 2.1.4: Did you merge data from all 6 satellite sensors? What spatial resolution did you use for merged product? If 1km, then OLCI data was "upsampled"?

Line 174: Keep consistent- change to 1 km

Line 175: Need citation

Line 176: Arc-seconds seems like a weird unit here.. can you convert to degrees or m?

Section 2.2.1: What did you use to create masks and "manually correct" discrepancies? Python? GIS?

Line 207: This needs a citation

Line 247: Are the equations in the paranthesis supposed to be exactly the same?

Line 249: What does SEM stand for here?

Lines 307-310: Do these sentences belong in the Methods?

Line 311: I don't think these figures are considered time-series? They are just snapshots, right?

Line 435: Change to [chla]- keep consistent

Figure 2: I wonder if labeling the islands on the map will help orient the readers?

Figure 4: What does "average or properties within the IME" mean?

**End of review**

---

## Author Comment (AC1)

We thank reviewer #1 for a thorough and constructive review. This review helped us to significantly improve the flow of the manuscript particularly the Methods section. All comments have been addressed as described below. Reviewer's comments are copied below and have been italicized, our responses are in normal black font, citation from the manuscript are in blue font,  text was deleted, and new sentences that were added in the revised version, in response to reviewer comments, are shown in red font text.

**Major comments**

*Introduction*

- *No major comments*

*Methods*

- *The Methods need substantial reorganizing and clarification:*

   o *More details should be added to the POLYMER atmospheric correction description. Why does it improve data recovery in areas impacted by glint and adjacency effect? What version did you use? Where was it downloaded from? What was it run on? What flags were used? What ancillary data was used?*

We added a statement in section 2.1.1

All analyses for this study were conducted using the University of Maine's high-performance Linux computing cluster following the processing pipeline shown in Fig. A1.

We modified paragraph 2.1 to include more information about POLYMER:

[Main text paragraph 2.1]:

2.1 Level-3 multi-satellite composites

Additionally, we applied the POLYMER atmospheric correction (Steinmetz et al., 2011) to further improve data recovery in areas impacted by glint and adjacency effect (e.g. close to shore and clouds. POLYMER is an atmospheric correction based on a spectral matching method to decompose the top of the atmosphere (TOA) signal into an atmospheric model and an ocean reflectance model. A three-term polynomial fit is used to model the atmospheric reflectance with the first term accounting for non-spectral scattering such as glint, and the last term accounting for adjacency effect from clouds and white surfaces (Steinmetz et al., 2011). By utilizing the entire TOA spectrum and accounting for adjacency effects and residual glint in its polynomial fit terms, this method improves the retrieval of high-quality data around clouds and from pixels affected by sun glint compared to standard atmospheric correction methods (Frouin et al., 2009, 2012). All analyses for this study were conducted using the University of Maine's highperformance Linux computing cluster following the processing pipeline shown in Fig. A1. .

The references Frouin et al., 2009, 2012 have been added to the reference list. We added the version, the download reference, information on ancillary data, and the computer system used in paragraph 2.1.1:

[Main text paragraph 2.1.1 – lines 85 - 86]:

We processed these L1A images into atmospherically corrected level-2 remote sensing reflectance ($R_{rs}$) data using the POLYMER algorithm (version v4.17beta2; Steinmetz 2023) and ancillary data from the European Centre for Medium-Range Weather Forecasts reanalysis model version 5 model (i.e. ERA5).

The reference associated with the POLYMER version used (version v4.17beta2; Steinmetz, 2023) from the HYGEOS GitHub has been added to the reference list.

We provided a direct reference for the flag application in the Readme file of POLYMER (POLYMER flags) that lists the flag recommendation. We prefer to keep the specificity of flags in the Readme file rather than listing the flags in the main text to avoid a long and hard to follow Methods section, but we can include it if necessary.

[Main text - line 86 – 87]:
We removed bad quality data pixels by applying the flags and recommendations of POLYMER (see reference POLYMER flags).

The reference associated with the readme file listing the flag recommendations (Steinmetz 2023a) from the HYGEOS GitHub has been added to the reference list.

> o *You should also be citing this paper as well: François Steinmetz and Didier Ramon "Sentinel-2 MSI and Sentinel-3 OLCI consistent ocean colour products using POLYMER", Proc. SPIE 10778, Remote Sensing of the Open and Coastal Ocean and Inland Waters, 107780E (30 October 2018);*
> *https://doi.org/10.1117/12.2500232*

We added a statement in paragraph 2.1, line 81:

[Main text paragraph 2.1]:

The POLYMER atmospheric correction was initially developed for the Medium Resolution Imaging Spectrometer sensor (MERIS) but was adapted to produce consistent ocean color products between MODIS, VIIRS, and OLCI sensors among others (Steinmetz et al. 2018).

The reference Steinmetz et al. (2018) have been added to the reference list.

○ *The headings in the Methods section seem disorganized to me. For Section 2.1 Level-3 Multi-satellite composites, you start with an intro paragraph and then have several subheadings. Consider merging the intro paragraph into the 2.1.1 section. The 2.1.2 In situ data sub-section seems out of place in this section, consider adding a new section just for in situ data and matchups. Perhaps this organization with just two sections?*

As suggested, we merged the opening paragraph and the subheadings 2.1.1 and 2.1.5 into section 2.1, however we decided to keep 2.1.4 separate because the merging into level-3 data described in this paragraph require the in situ sampling to be introduced before. Therefore, we have removed the sub-heading as suggested and kept only 3 sections explaining the method to create level-3 merged products:

- 2.1 Level-3 satellite products computation:
  This section includes the opening paragraph. 2.1.1 and 2.1.5

- 2.2 In situ data and matchups
  This section includes section 2.1.2 and 2.1.3

- 2.3 Level-3 multi-satellite products merging
  This section includes 2.1.4

We also noticed $\Sigma[Chla]_{IME}$ (section 2.1.1 lines 90 – 93) should be described after the method to detect the IME zones (section 2.2.2). We moved "We computed surface-integrated [Chla] as a metric for two dimensional phytoplankton biomass in metric tons of Chla per depth meter (mt.m−1) by summing the [Chla] of each pixel within a predefined zone (i.e. here, the zone influenced by IME) multiplied by the area of that pixel:

$$\sum[Chla]_{IME} = \sum_{n=1}^{N_{pixel_{IME}}} [Chla]_n \times area_{pixel_n}$$
"

from section 2.1.1 lines 90 – 93 to section 2.2.2 line 245.

○ *The description of running POLYMER and l2gen should be in same paragraph/section. Right now, you have text on l2gen in the in situ and satellite matchups section which seems out of place.*

We moved "For comparison, we also generated the standard NASA Rrs using the atmospheric correction of SeaDAS (i.e. "l2gen") using the Ocean Color processor (OCSSW) V2022.3. We then estimated [Chla] from these Rrs using the same blended CI-OCx algorithm (i.e. chlor_a; Hu et al., 2019) and the simple OCx algorithm (i.e. chl_ocx; O'Reilly and Werdell, 2019)." into section "2.1 Level-2 satellite products computation".

○ *In Section 2.1.1, there is no description of how data was processed to Level-3 format. It seems to stop at L2.*

We have addressed this issue while re-organizing the method sections. The level-2 data production is now explained in section "2.1 Level-2 satellite products computation" and the merging of these level-2 data into level-3 products is described into "2.3 Level-3 multi-satellite products merging"

      o   *Figure 1 seems okay in the Methods because it is a figure of the workflow. However, should Figures 2-3 be in the "Assessment" section since it is showing the results of the workflow? I don't think you should be referencing results figures in the Methods, save that for the Results (or "Assessment").*

We decided to include Figures 2 and 3 into the method to help understand how IME zones are not strictly developing around individual islands but instead encompassing island groups. We believe these two figures provide a good illustration justifying the strategy to detect IME around neighboring islands in section 2.2.3.

*Assessment*

- *Some of the text in the Assessment section would better belong in the Methods such as the description of merging and binning and the chl iteration step size*

We have moved the sentence about binning Lines 307-310 to the method section 2.3 such as suggested in the minor comment below.

We acknowledge the "Assessment" section still contains some methodological details including merging, binning, and the chl iteration step size however all these details are first presented in the method section. In the "Assessment" section, we assess our new approach, including the impact of merging, binning, and changing the iteration step size to demonstrate how these changes enhance IME detection using the algorithm developed in this study.

We have modified the sentence introducing the changes applied to the method from Messié et al. (2022) in the method section (line 206) to clarify that all the following statements describe the method updates developed in this study:

"We therefore extended the method proposed by Messié et al. (2022) by adding another set of detection protocols, here called step 2 and step 3."

**Minor comments**

- *The title on the preprint PDF is different than what is in the system.*

Thank you for pointing this issue, we did not updated the title in the system with the last version, we will make sure to correct this mistake.

- *Line 5: Consider adding a after chlorophyll. Same on Line 20.*

We added "a" after chlorophyll line 5 and 20.

- *Line 8: Define POLYMER*

We would like to thank the reviewer for this suggestion. We have added a detailed description of POLYMER in section 2.1

- *Line 18: The way this sentence is written makes it seem like "their wake downstream.." refers to the winds and currents. Reword to make this more clear.*

We have reworded the sentence line 18 to improve clarity:

As winds and currents interact with island topography, they induce mesoscale processes (i.e. local upwelling, eddies) that form at the downstream wake of islands .

- *Line 28: Consider adding the citation to the end of this sentence.*

Since these two sentences refer to the same citation, we have moved the citation (Doty and Oguri, 1956) from line 29 to line 30.

- *Line 35: Consider changing "They" to "The authors"*

We have changed "They" by "The authors" line 35.

- *Line 77: Should define these satellite mission acronyms*

We have defined these satellite mission acronyms line 77.

- *Line 79: More information on the POLYMER atmospheric correction scheme should be included here. See major comment above.*

We have included information on the POLYMER atmospheric correction in this section.

- *Line 84: Include the time frame of data collected. What is "all"?*

We have added "consisted in six-month long time-series of satellite data" in this paragraph when we re-worked part of this section to address the major comment regarding method reorganization.

- *Line 84: Why did you download L1A data instead of L1B? Review differences here: https://oceancolor.gsfc.nasa.gov/resources/docs/productlevels/#:~:text=Level%201B%20 data%20are%20Level,had%20instrument%2Fradiometric%20c alibrations%20applied.&text=Level%202%20data%20consist%20of,the%20source%20 Level%2 01%20data.*

We chose to download L1A instead of L1B data because the Python utility "getOC" used to batch download all satellite data only support L1A and L2 data for NASA satellite. Both L1A and L1B data can be processed using the POLYMER atmospheric correction.

- *Line 85: What Copernicus repository? Provide link(s).*

We downloaded data via the Copernicus Data Space Catalogue API. We have changed the sentence added a link for both NASA Ocean Color API and Copernicus API.

"We downloaded level-1 (L1A) top-of-the-atmosphere radiance of MODIS-Aqua, MODIS-Terra, VIIRS-SNPP, and VIIRS-JPSS1 via NASA's common metadata repository application programming interface (CMR API), and the resampled 1 km spatial resolution OLCI-S3a and OLCI-S3b data via the Copernicus Data Space Catalogue API using the Python download utility "getOC" (getOC GitHub: Haëntjens and Bourdin, 2017)all MODIS and VIIRS level-1 (L1A) images in the vicinity of islands of interest from the Ocean Color repository, and OLCI level-1 images from the Copernicus repository."

- *Line 88: What did you use to project the satellite data onto a plate-carre reference grid using NN interpolation?*

We used the Python SciPy library to project satellite data onto a plate-carre reference grid using NN interpolation and added this information in section 2.1.

"Subsequently, we projected each satellite image of the "study dataset" onto the same equally spaced 1 km spatial resolution plate-carré reference grid specific to each studied region of interest using nearest-neighbor interpolations from Python's SciPy library."

- *Line 91: Confused on how this is surface-integrated chla when you're just summing chla concentration in each pixels by the area? Where does depth come into play?*

We renamed this parameter "surface-area integrated [Chla]" to clarify that it corresponds to [Chla] integrated over the surface area of IME and BO zones and not depth:
"We computed surface-area integrated [Chla] as a proxy for surface phytoplankton biomass integrated over entire IME and BO zones in two-dimensional metric for two dimensional phytoplankton biomass in metric tons of chlorophyll *a* per depth meter (mt.m−1) by summing the [Chla] of each pixel within IME and BO zones a predefined zone (i.e. here, the zone influenced by IME) multiplied by the area of that pixel"

We changed "surface-integrated [Chla]" with "surface-are integrated [Chla]" throughout the text.

This description is now located in section formerly named "2.2.2 IME contour delineation" and now "2.4.2 IME contour delineation", after the description of the IME and BO zones which improves clarity.

- *Line 96: Not sure you need to hyphenate hyperspectral*

The hyphen was deleted in "hyperspectral".

- *Line 110: Capitalize Python*

We capitalized "Python".

- *Line 112: You describe how all satellite data is processed to Level-3 using same scheme as aforementioned but this was never described.. You don't introduce the terms reprojecting, nudging, or merging until now. What is nudging?*

This issue was addressed while re-organizing the method section following the major comment above. "Nudging" in this case means "adjusting" or "calibrating" satellite data to in situ data. Section 2.3 now explains the nudging and merging methods before it is mentioned elsewhere.

- *Line 114: OCSSW stands for Ocean Color Science Software*

The OCSSW acronym was defined as suggested.

- *Line 119: Consider adding the satellite overpass times for each sensor. How do they match up with the 10:30am local time for in situ data collection?*

We have added each satellite overpass time for reference in section 2.2. Satellite data were not matched with HPLC sampled at 10:30 local time. As explained in section 2.1.2 correlations between $a_p$ and total chlorophyll $a$ concentration were used to estimate [Chla] from continuous measurements of $a_p$, and as explained in section 2.1.3 satellite data were matched with these continuous estimates of [Chla].

- *Line 120: The sentence about recommended Level-2 masks needs a citation. Masks or flags? Did you use recommended L2 or L3 flags?*
  *https://oceancolor.gsfc.nasa.gov/resources/atbd/ocl2flags/*

We applied the L2 default flags. We have clarified this sentence and moved it to section 2.1 following the method section reorganization: "For comparison, we also generated the standard NASA Rrs using the atmospheric correction of SeaDAS (i.e. "l2gen") using the Ocean Color Science Software (OCSSW) V2022.3, on which we applied the Level-2 default flags (NASA OBPG flags)."

The parenthesis (NASA OBPG flags) refers to the level-2 default flags webpage

- *Line 120: Are you working with Rrs or nLw? Are these both included when running l2gen and POLYMER?*

We have corrected this sentence we computed median coefficients of variation or Rrs not nLw to check for match-ups validity. We only generated Rrs using l2gen and POLYMER for this study. Note that Rrs or nLw can be used interchangeably to check for match-up validity using their coefficient of variance because nLw = Rrs × $F_0$ where $F_0$ is the extra-terrestrial solar flux at the time of observation.

- *Line 136: What is GlobColour?*

GlobColour is the ocean color merging processor distributed by Copernicus. We added "Copernicus' multi-satellite Global Ocean Colour processor (i.e. GlobColour)" to explain what GlobColour stands for. More details about the GlobColour can be found in the work cited in this sentence (Garnesson et al., 2019).

- *Line 138: Why would this described merging strategy require simulation of 510 nm band? Line 162: This sentence should have a citation*

By "*simulation of 510 nm band*" we meant band shifting procedure to approximate Rrs for bands not available for all sensors before merging Rrs. We have clarified this sentence:

"This method offers two important advantages; (1) it does not require any band shifting procedure to merge Rrs between sensors with different spectral bands  and (2) it benefits from sensor-specific algorithm coefficients that account for variability in Rrs across sensors to produce consistent products (Garnesson et al., 2019)."

- *Line 168: Did you use the 300m spatial resolution of OLCI? Section 2.1.4: Did you merge data from all 6 satellite sensors? What spatial resolution did you use for merged product? If 1km, then OLCI data was "upsampled"?*

We used resampled OLCI products distributed by Copernicus to start at the same nominal spatial resolution than MODIS and VIIRS L2 products. We added this information in section 2.1:

"We downloaded level-1 (L1A) top-of-the-atmosphere radiance of MODIS-Aqua, MODIS-Terra, VIIRS-SNPP, and VIIRS-JPSS1 from the Ocean Color repository, and the resampled 1 km spatial resolution OLCI-S3a and OLCI-S3b data from the Copernicus Data Space repository using the Python download utility "getOC" (getOC GitHub: Haëntjens and Bourdin, 2017 )."

- *Line 174: Keep consistent- change to 1 km*

We changed "one kilometer" to "1 km" line 174 and throughout the text and captions

- *Line 175: Need citation*

We added the citation for the GEBCO 2022 database used.

- *Line 176: Arc-seconds seems like a weird unit here.. can you convert to degrees or m? Section 2.2.1: What did you use to create masks and "manually correct" discrepancies? Python? GIS?*

Arc seconds is the unit of angular measurement at which GEBCO is distributed (see associated citation). It corresponds to 463m at the equator such as mentioned in the same parenthesis line 176: "(i.e. 15 arc-seconds corresponding to 463 m at the equator)."

- *Line 207: This needs a citation*

The citation for the global ocean ensemble physics reanalysis products was added.

- *Line 247: Are the equations in the paranthesis supposed to be exactly the same?*

The equations in the parenthesis are not the same, one represents delta chlorophyll concentration ($\Delta[\text{Chla}]_{\text{IME}_T-\text{BO}_T}$) while the other represents delta surface-area integrated chlorophyll ($\Delta\Sigma[\text{Chla}]_{\text{IME}_T-\text{BO}_T}$).

- *Line 249: What does SEM stand for here?*

SEM stands for standard error of the mean and was indeed only defined in the table of notation (Table 1). We have defined SEM at its first occurrence line 154

"We propagated errors associated with [Chla] estimation, nudging, and merging throughout each step to represent the final [Chla] uncertainty denoted as the standard error of mean (i.e. SEM) of the merged product ($SEM^f_{[Chla]_{IME}}$ ; see appendix B)."

- *Lines 307-310: Do these sentences belong in the Methods?*

Yes, the sentence starting line 307 belongs in the Methods, we moved it to section 2.1.4 and adjusted the previous two sentences accordingly:
"Time-series of 8-day  periods were the smallest temporal binning we could achieve to recover nearly full satellite images in all the studied regions for six-month long time-series. Before computing the  merged products of a given 8-day period and a given region, we grouped all re-projected level-2 images and removed outliers (see appendix C). To minimize the weight of outliers on the end level-3 products, the binning was performed with medians instead of averages."

- *Line 311: I don't think these figures are considered time-series? They are just snapshots, right?*

Indeed, these figures are snapshots. We added the reference for the time-series from the supplementary material and modified the sentence accordingly:
"Time-series of remote sensing maps (Bourdin, 2024a) and their snapshots (Fig. 2, Fig. 3) reveal the complexity of currents around islands and the rather chaotic advection patterns of IME into the open-ocean and between islands."

- *Line 435: Change to [chla]- keep consistent*

We chose to use chlorophyll *a* instead of [Chla] in this specific case because [Chla] refers to chlorophyll *a* concentration but this sentence is about absolute accumulation of chlorophyll *a*, not the concentration of chlorophyll *a*.
"This dynamic IME detection method permitted tracking in time the accumulation of chlorophyll *a* standing stock in surface waters, which suggested frequent temporal increases in phytoplankton biomass in addition to the spatial increase in phytoplankton biomass already detected around islands."

- *Figure 2: I wonder if labeling the islands on the map will help orient the readers?*

We thank the reviewer for this suggestion, we added labels on the map depicting the main archipelagoes.

- *Figure 4: What does "average or properties within the IME" mean?*

We acknowledge "average of properties" in Figure 4 and 5 captions are confusing. We modified the captions of Figure 4, 5, E1, and E2 to present separately each panel of the figures.

Please find below the final draft of the first 3 sections of the methods we have re-organized:

[revised manuscript text omitted]

**End of review**

---

## Author Comment (AC2)

We thank reviewer #2 for raising an important concern regarding the use of MODIS-Terra data into a multi-satellite merged product. We added a statement justifying this choice based on correlations with in situ data. Reviewer's comments are copied below and have been italicized, our responses are normal black font, citation from the manuscript are in blue font,  text was deleted, and new sentences that were added in the revised version, in response to reviewer comments, are shown in red font text.

*This contribution investigates the Island Mass Effect (IME) in the Pacific Ocean, highlighting its potential significant impact on biogeochemical processes in oligotrophic waters surrounding remote islands and atolls. The study expands on the limitations of traditional remote sensing approaches that rely on L3 data with maximum resolution above 4km, which often fail to capture the full extent of sub-mesoscale and short-term temporal variations. It proposes an alternative enhanced approach that merges multi-sensor satellite data at a higher spatial resolution and integrates modelled surface currents to dynamically track chlorophyll enhancements associated with IME. The methodology is applied to four South Pacific island groups, suggesting that the ecological influence of IME may be larger than previously recognized, with important implications for broader ocean ecological studies.*

**Strengths of the Contribution**

1. ***Innovative Methodology:*** *The research introduces a layered and carefully crafted methodology, combining ocean color data processing with modeled dynamic tracking of chlorophyll patches and filaments, as well as in situ calibration and validation measurements. This custom approach, it is proposed, addresses the limitations of legacy IME detection algorithms and significantly improves the spatial resolution of feature retrieval.*

2. ***Broader Implications:*** *By showcasing the potential broader ecological influence of IME, the study advances our understanding of global oceanic biogeochemistry. It provides a valuable foundation for investigating these processes in strongly stratified systems, such as the tropical western Pacific, where islands and submerged topography, as defined here, may cause significant perturbations that lead to enhanced productivity. This is particularly well articulated in L377, where the authors underscore the importance of IME. These regional results should be further evaluated in the context of global biogeochemical cycles, as well as in other island systems where background ocean biogeochemistry creates more eutrophic conditions (e.g., Galapagos, Ascension, Azores, etc.).*

**Areas for Improvement**

*While the paper is a worthy contribution, some areas could benefit from refinement:*

1. ***General study design (sensor choice justification):*** *The study effectively combines data from multiple sensors, including MODIS Terra, to detect changes in chlorophyll associated with the Island Mass Effect. While this approach is justified given the study's focus on detecting relative changes rather than establishing absolute chlorophyll*

*concentrations, the authors should include a brief discussion of the known issues with MODIS Terra. Specifically, acknowledging its calibration challenges, and potential limitations for climate-quality data would enhance the transparency and robustness of the methodology. This acknowledgment would reassure readers that these factors have been considered and appropriately addressed in the study's design.*

We would like to thank the reviewer for raising this concern regarding calibration challenges of MODIS-Terra. We added a short statement at the end of section "2.1.3 In situ and satellite match-ups":

Despite the well-documented degradation of the MODIS sensor onboard the Terra satellite and its potential impact on climate studies (Lyapustin et al., 2014; Xiong et al., 2019; Xiong and Butler, 2020), our analysis found no significant indication of reduced data quality in [Chla] estimates derived from MODIS-Terra Rrs. Correlations between in situ [Chla] and MODIS-Terra-derived [Chla] showed performance metrics ($R^2$, nRMSE, slope, and intercept) comparable to those of other satellite sensors included in this study (Table B1 and Fig. B2 b, c, and d). These findings suggest that the extensive correction and calibration efforts applied to MODIS-Terra data effectively mitigate the impacts of solar diffuser degradation, changes in scan mirror reflectance, and increased polarization sensitivity (Lyapustin et al., 2014). As a result, MODIS-Terra data can be reliably incorporated into the multi-satellite merged product used in this study.

The references Lyapustin et al., 2014; Xiong et al., 2019; Xiong and Butler, 2020 have been added to the reference list.

2. ***Section 3 (Assessment):*** *The results section contains substantial material (e.g., L315-320) that is methodological in nature. For clarity and better flow, this information should be moved to the methods section. This reorganization will help strengthen the distinction between methodology and results.*

The sentence starting line 307 belongs in the Methods, we moved it to section 2.1.4 and adjusted the previous two sentences accordingly:
"Time-series of 8-day  periods were the smallest temporal binning we could achieve to recover nearly full satellite images in all the studied regions for six-month long time-series. Before computing the  merged products of a given 8-day period and a given region, we grouped all re-projected level-2 images and removed outliers (see appendix C). To minimize the weight of outliers on the end level-3 products, the binning was performed with medians instead of averages."

We acknowledge the "Assessment" section still contains some methodological details including merging, binning, and the chl iteration step size however all these details are first presented in the method section. In the "Assessment" section, we assess our new approach, including the impact of merging, binning, and changing the iteration step size to demonstrate how these changes enhance IME detection using the algorithm developed in this study.

We have modified the sentence introducing the changes applied to the method from Messié et al. (2022) in the method section (line 206) to clarify that all the following statements describe the method updates developed in this study:

"We therefore extended the method proposed by Messié et al. (2022) by adding another set of detection protocols, here called step 2 and step 3."